# Hippocampal place cell remapping occurs with memory storage of aversive experiences

**Garrett J Blair[1]\*[†], Changliang Guo[2,3,4], Shiyun Wang[1], Michael S Fanselow[1,4], Peyman Golshani[2,3,4], Daniel Aharoni[2,3,4], Hugh T Blair[1,4]\***

[1]Department of Psychology, UCLA, Los Angeles, United States; [2]David Geffen School of Medicine, University of California Los Angeles, Los Angeles, United States; [3]Department of Neurology, David Geffen School of Medicine, University of California, Los Angeles, Los Angeles, United States; [4]Integrative Center for Learning and Memory, University of California, Los Angeles, Los Angeles, United States

**Abstract** Aversive stimuli can cause hippocampal place cells to remap their firing fields, but it is not known whether remapping plays a role in storing memories of aversive experiences. Here, we addressed this question by performing in vivo calcium imaging of CA1 place cells in freely behaving rats (n = 14). Rats were first trained to prefer a short path over a long path for obtaining food reward, then trained to avoid the short path by delivering a mild footshock. Remapping was assessed by comparing place cell population vector similarity before acquisition versus after extinction of avoidance. Some rats received shock after systemic injections of the amnestic drug scopolamine at a dose (1 mg/kg) that impaired avoidance learning but spared spatial tuning and shock-evoked responses of CA1 neurons. Place cells remapped significantly more following remembered than forgotten shocks (drug-free versus scopolamine conditions); shock-induced remapping did not cause place fields to migrate toward or away from the shocked location and was similarly prevalent in cells that were responsive versus non-responsive to shocks. When rats were exposed to a neutral barrier rather than aversive shock, place cells remapped significantly less in response to the barrier. We conclude that place cell remapping occurs in response to events that are remembered rather than merely perceived and forgotten, suggesting that reorganization of hippocampal population codes may play a role in storing memories for aversive events.

## Editor's evaluation

This paper describes important results obtained from multi-cellular imaging of CA1 cells using large-field-of-view miniscopes in rats performing a shock avoidance task. By exploiting behavioral (barriers) and pharmacological (scopolamine) manipulations the authors provide convincing results on cell remapping dynamics during aversive learning. This work will be of interest for the neuroscience community by setting new methodological standards and providing data for across species comparisons.

## Introduction

The rodent hippocampus contains place cells that fire selectively during visits to specific locations in space (*O'Keefe and Dostrovsky, 1971*). When place cells were first discovered, they were hypothesized to encode 'cognitive maps' that store stable long-lasting memories for the spatial geometry of familiar environments (*O'Keefe and Nadel, 1978*). Supporting this view, electrophysiology studies

**\*For correspondence:**
gblair@ucla.edu (GJB);
tadblair@ucla.edu (HTB)

**Present address:** [†]Center for Neural Science, New York University, New York, United States

**Competing interest:** The authors declare that no competing interests exist.

**eLife digest** The human brain is able to remember experiences that occurred at specific places and times, such as a birthday party held at a particular restaurant. A part of the brain known as the hippocampus helps to store these episodic memories, but how exactly is not fully understood.

Within the hippocampus are specialized neurons known as place cells which 'label' locations with unique patterns of brain activity. When we revisit a place, such as the restaurant, place cells recall the stored pattern of brain activity allowing us to recognize the familiar location.

It has been shown that a new negative experience at a familiar place – for example, if we went back to the restaurant and had a terrible meal – triggers place cells to update the brain activity label associated with the location. However, it remains uncertain whether this re-labelling assists in storing the memory of the unpleasant experience.

To investigate, Blair et al. used a technique known as calcium imaging to monitor place cells in the hippocampus of freely moving rats. The rats were given a new experience – a mild foot shock – at a previously explored location. Tiny cameras attached to their heads were then used to record the activity of hundreds of place cells before and after the shock.

Initially, the rats remembered the aversive experience and avoided the location where they had been shocked. Over time, the rats began to return to the location; however, their place cells displayed different patterns of activity compared to their previous visits before the shock.

To test whether this change in place cell activity corresponded with new memories, another group of rats were administered a mild amnesia-inducing drug before the shock, causing them to forget the experience. These rats did not avoid the shock site or show any changes in place cell activity when they revisited it.

These findings imply that new events cause place cells to alter their 'label' for a location only if the event is remembered, not if it is forgotten. This indicates that alterations in place cell activity patterns may play a role in storing memories of unpleasant experiences. Having a better understanding of how episodic memories are stored could lead to better treatments for diseases that impair memory, such as Alzheimer's disease and age-related dementia.

showed that place cells can retain stable positional tuning across repeated visits to the same environment (*Lever et al., 2002*). However, methodological limitations of single-unit electrophysiology made it challenging to consistently record the same population of place cells across long time spans, and thus to investigate the duration of time over which place cell tuning properties remained stable.

Recent advances in in vivo calcium imaging have made it possible to perform longitudinal studies of place cell tuning over days or weeks of experience. Such studies suggest that place cells in mice (*Ziv et al., 2013*; *Cai et al., 2016*; *Kinsky et al., 2018*; *Keinath et al., 2022*) and to a lesser extent in rats (*Wirtshafter and Disterhoft, 2022*) can change their spatial tuning properties (i.e., 'remap') with the mere passage time (see also *Mankin et al., 2012*). This phenomenon suggests that place cells may not simply encode spatial maps of environmental geometry, but also encode mnemonic information about accumulated past experiences that have occurred within an environment (*Leutgeb et al., 2005*; *Colgin et al., 2008*; *Mankin et al., 2012*; *Ziv et al., 2013*; *Cai et al., 2016*; *Sanders et al., 2020*). If so, then the rate at which place cells remap over time within a given environment may be proportional to the rate at which an animal stores new episodic memories of events that are linked to that environment.

If the rate of place remapping reflects the rate of hippocampal memory storage, then motivationally significant events (such as encounters with appetitive or aversive stimuli) that are preferentially stored to episodic memory might be expected to accelerate the rate of place cell remapping within environments where such events occur. Consistent with this view, studies have shown that place cells remap in response to behaviorally significant events (*Breese et al., 1989*; *Bostock et al., 1991*; *Hollup et al., 2001*; *Colgin et al., 2008*; *Dupret et al., 2010*; *Alvernhe et al., 2011*; *Mamad et al., 2017*). Numerous studies have shown that when an aversive stimulus is encountered in an environment, place cells respond by remapping their firing fields in that environment (*Moita et al., 2003*; *Moita et al., 2004*; *Wang et al., 2012*; *Wang et al., 2015*; *Kim et al., 2015*; *Mamad et al., 2019*;

*Schuette et al., 2020*; *Ormond et al., 2023*), but it is not well understood whether such remapping plays a role in storing memories of aversive experiences.

Here, we investigated the role of place cell remapping in memory by using the MiniLFOV (*Guo et al., 2023*), a recently developed large-field-of-view version of the UCLA Miniscope (*Cai et al., 2016*; *Aharoni and Hoogland, 2019*), to image population activity of CA1 place cells in freely behaving rats as they acquired and extinguished a spatial avoidance learning task. The task was performed on a rectangular maze where rats ran stereotyped linear trajectories, allowing the speed and direction of running behavior to be closely matched during sessions conducted before acquisition versus after extinction of avoidance learning. In agreement with prior studies, we observed that avoidance learning induced place cell remapping in an environment where footshock was encountered compared with control conditions where (1) rats encountered a neutral barrier instead of an aversive footshock or (2) rats received footshock after injections of the muscarinic ACh receptor (mAChR) antagonist scopolamine at a dose (1 mg/kg) that disrupted long-term retention of avoidance without acutely impairing immediate avoidance or spatial tuning of place cells. Remapping after avoidance learning was similarly prevalent in subpopulations of shock-responsive and non-responsive place cells, and in contrast with some prior findings (*Mamad et al., 2019*), remapping did not alter the spatial distribution of place fields on the maze (i.e., there was no net migration of place fields toward or away from the shocked location). These findings add to a large body of evidence showing that place cell remapping occurs in response to aversive events, and also provide new evidence that remapping occurs preferentially under conditions where aversive encounters are remembered rather than forgotten, consistent with the hypothesis that place cell remapping may play a role in storing memories of aversive encounters.

## Results

Male (n = 7) and female (n = 8) Long–Evans rats underwent survival surgery to inject 1.2 uL of AAV9-Syn-GCamp7s (*AddGene*) in the hippocampal CA1 layer; ~2 wk later, a 1.8-mm-diameter GRIN lens stack (~0.5 pitch) was unilaterally implanted to image CA1 neurons. A baseplate was attached to the skull for mounting the miniscope, and then rats began a regimen of running one 15 min session per 48 hr on a 250 × 125 cm rectangular maze (*Figure 1A*). Rats earned 20 mg chocolate pellets by alternating between two rewarded corners. On each trial, rats were free to choose a direct short path (250 cm) or an indirect long path (500 cm) to reach the next reward. During early sessions, rats learned to prefer the short over the long path (*Figure 1C and D*). Upon reaching criterion for short path preference (>2 rewards/min, and twice as many short paths to long paths), rats began receiving drug-free shock, scopolamine shock, or barrier training sessions (*Figure 1B*).

### Acquisition and extinction of avoidance learning

During shock training sessions (drug-free or scopolamine), rats performed the standard alternation task for 10 min. Then a 1.0 mA scrambled current was switched on to electrify grid bars in a 50 cm segment of the short path's center (*Figure 1A*). Identical grid bars spanned the full length of the long and short paths, so there were no local cues to indicate the shock zone's location. After encountering the shock zone 1–2 times, rats subsequently avoided the short path during the final 5 min of the session (*Figure 1E*). In the scopolamine shock condition, lens-implanted rats received 1.0 mg/kg scopolamine via intraperitoneal injection 30 min prior to the start of the shock session. In the drug-free condition, lens-implanted rats received no injection prior to the start of the shock session. To control for effects of the injection procedure, a separate cohort of rats without lens implants was given injections of saline or scopolamine prior to shock training; in animals without lens implants, scopolamine-injected rats showed impaired avoidance learning compared with saline-injected rats that was similar to the impairment seen between scopolamine-injected and drug-free (non-injected) rats with lens implants (*Figure 1—figure supplement 1B and C*), indicating that the injection procedure itself was not responsible for effects induced by scopolamine. In the barrier control condition, the maze was not electrified; instead, a clear plexiglass barrier was placed in the center of the short path, forcing the rat to take the long path during the final 5 min of the session.

During the first 10 m of training on scopolamine (before shock delivery), 6/10 rats (three males, three females) fell below the criterion for preferring the short path (compared with 0/14 drug-free rats). These rats earned a median of five short path rewards/min during the drug-free session given

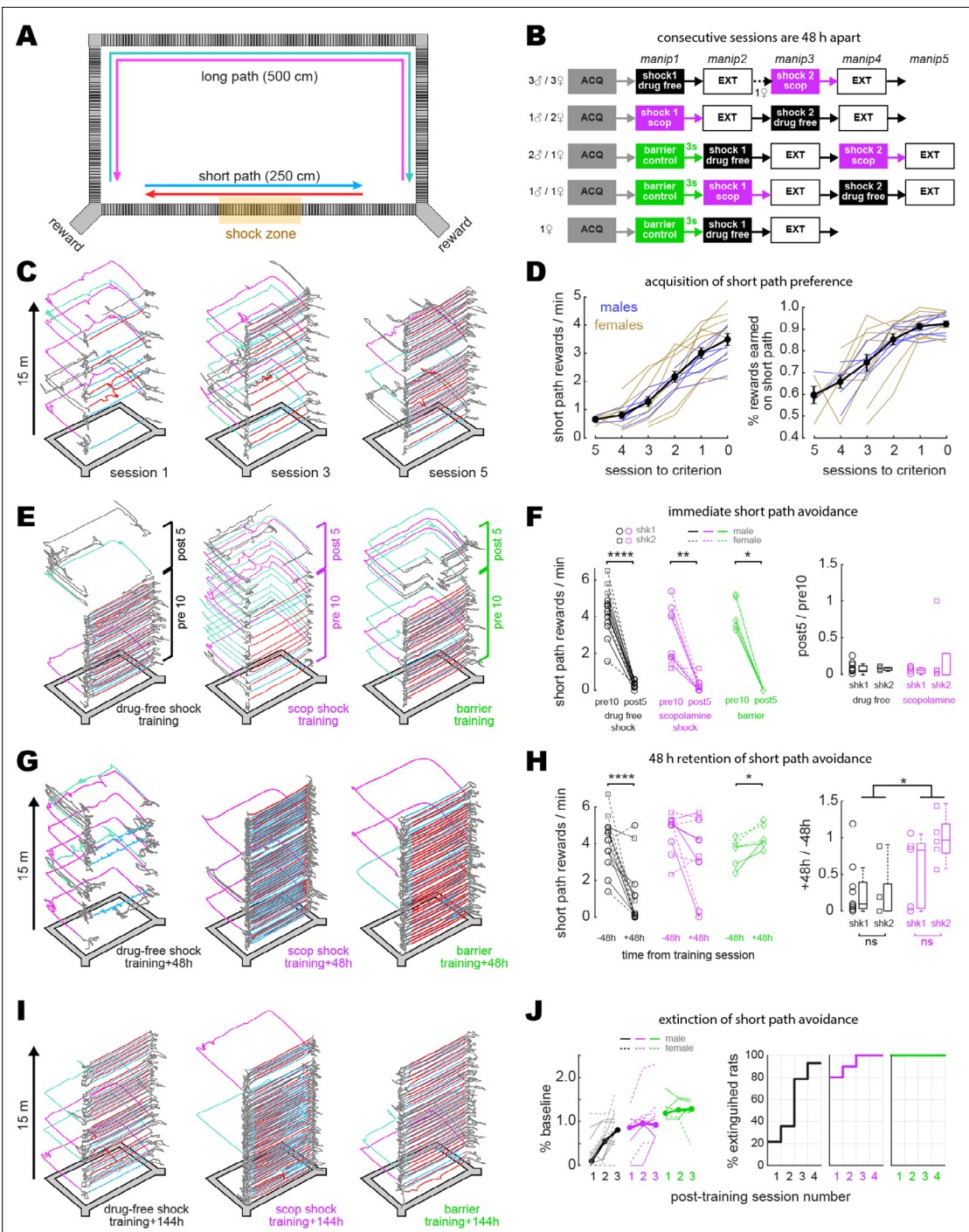

**Figure 1.** Disruption of aversive avoidance acquisition by scopolamine. (**A**) Overhead view of the rectangular maze. (**B**) Order of treatment for five subgroups of rats in the study. The number of sessions in ACQ and EXT boxes varied per rat, depending on how many sessions it took individual subjects to reach behavioral criteria. (**C**) Example trajectories from sessions 1, 3, and 5 during initial acquisition of short path preference. (**D**) Acquisition curves for short path preference; criterion was reached on session 0. Black lines show mean +/- 1 standard deviation for the session (**E**) Example maze trajectories from training sessions. (**F**) Left: rewards/min earned on the short path during the first 10 min (pre10) versus last 5 min (post5) of training sessions; symbols indicate whether each rat received its first (shk1) or second (shk2) shock during a session. Right: boxplots show post-shock reduction in short path rewards/min (post5/pre10). (**G**) Example maze trajectories 48 hr after training. (**H**) Left: rewards/min earned on the short path during drug-free sessions given 48 hr before (−48 hr) vs. after (+48 hr) training. Right: boxplots show 48 hr retention of shock avoidance (+48 hr/ −48 hr). (**I**) Example maze trajectories 144 hr after training. (**J**) Left: extinction curves over the first three post-extinction sessions; number of rewards earned on the short path is measured as a percentage of baseline (48 hr before training; thin lines: individual rats, thick lines: per session median). Right: cumulative distributions

*Figure 1 continued on next page*

*Figure 1 continued*

show percentage of rats in each training condition meeting extinction criterion (≥2 rewards/min on the short path, and at least twice as many short path rewards as long path rewards) at 1, 2, 3, or 4 d after training. *p<0.05, **p<0.01, ****p<0.0001.

The online version of this article includes the following figure supplement(s) for figure 1:

**Figure supplement 1.** Avoidance learning in male versus female and scopolamine versus saline injected rats.

48 hr before training, but only 1.9 short path rewards/min during the first 10 min (before shock) of their training session on scopolamine (*Figure 1F*, 'scopolamine shock pre10'; Wilcoxon test, p=0.0098). Thus, prior to shock delivery, scopolamine impaired expression of the rats' learned preference for taking the short path (*Figure 1E*, middle).

During the training session, rats earned fewer rewards/min on the short path after than before grid electrification (*Figure 1F*) in both the drug-free (n = 14, signed-rank test, p=1.2e⁻⁴) and scopolamine (n = 10, signed-rank test, p=0.004) conditions. Rats in the barrier condition (n = 6) were likewise forced to stop taking the short path after it was blocked by the barrier (*Figure 1F*, left panel). Rats in the drug-free shock condition showed long-term retention of avoidance (*Figure 1G*, left), earning fewer rewards/min on the short path 48 hr after than 48 hr before training (*Figure 1H*, Wilcoxon test, p=3.7e⁻⁴). By contrast, most rats trained on scopolamine showed similar short path reward rates during drug-free sessions given 48 hr before and after training (*Figure 1H*, Wilcoxon test, p=0.125), and thus failed to exhibit long-term retention of avoidance (*Figure 1G*, middle). When tested 48 hr after training, rats shocked on scopolamine showed less avoidance of the short path than rats shocked drug-free (rank-sum test: p=0.0235; *Figure 1H*). Hence, pre-training scopolamine injections impaired aversive learning, consistent with prior studies (*Decker et al., 1990*; *Anagnostaras et al., 1999*; *Green et al., 2005*).

## Between- versus within-session place cell stability

A large-field-of-view version of the UCLA Miniscope ('MiniLFOV'; *Guo et al., 2023*) was used to image neurons in the hippocampal CA1 layer through the implanted GRIN lens during maze sessions (*Figure 2A*). Although the GRIN lens caused some flattening and compression of stratum oriens (*Figure 2—figure supplement 1*), this did not appear to cause significant disruptions of CA1 activity since spatial tuning properties of place cells were stable and intact (see below).

Calcium activity was analyzed only during traversals of the short path since the long path was under-sampled after rats had learned to prefer the short path. Analysis was further restricted to *beeline trials* during which the rat ran directly from one reward zone to the other on the short path, without any pause or change in direction (*Figure 2B*, *Video 1*); this assured that imaging data came from periods of active locomotion when place cells exhibit reliable spatial tuning, and that the rats' behavior during imaging was similar across all experimental sessions and conditions. To further control for possible confounding effects of behavior differences between sessions (see *Figure 2—figure supplement 2*), beeline trials were randomly subsampled using an algorithm (see 'Methods') that minimized trial count and running speed differences between the drug-free and scopolamine conditions (*Figure 2C*). Spatial tuning was then analyzed and compared during these behaviorally homogeneous trials. To analyze spatial tuning, the short path was subdivided into 23 spatial bins (each 10.8 cm wide). Two spatial tuning curves (one per running direction) were derived for each neuron. Cell activity was measured in units of *active frames per second* (Af/s), defined as the mean number of imaging frames per second during which a neuron generated at least one inferred spike. A neuron was classified as a place cell if its LR or RL tuning curve (or both) met defined criteria for minimum Af/s, spatial selectivity, and spatial stability within a session (see 'Methods'). About 1/3 of place cell tuning curves met the criteria for spatial selectivity only in the LR direction, 1/3 only in the RL direction, and 1/3 in both directions (*Figure 2E*). It should be noted that since we used a ubiquitous synapsin promoter to drive expression of GcaMP7s, some neurons classified as place cells might have been CA1 interneurons rather than pyramidal cells.

To analyze acute effects of scopolamine on CA1 place cell activity, data from drug-free or scopol-amine shock sessions was compared against data from pre-training baseline sessions (always drug-free) given 48 hr earlier. The drug-free condition only included rats (n = 9) that retained avoidance of the short path 48 hr aftershock training, whereas the scopolamine condition only included rats (n =

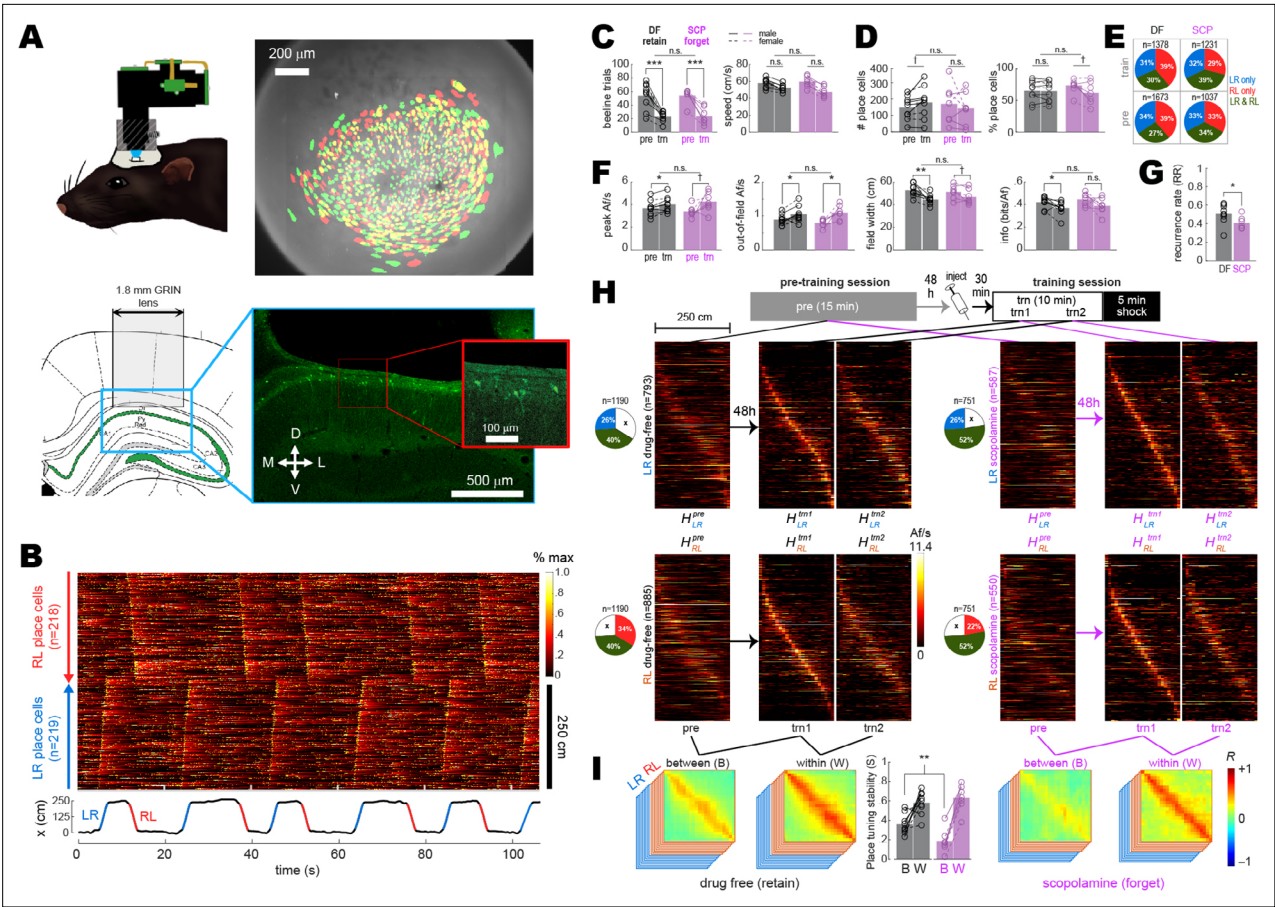

**Figure 2.** Between- versus within-session population coding. (**A**) Upper left: illustration of rat wearing MiniLFOV. Upper right: cell contours identified during pre-training (green) and training (red) sessions in one rat; regions of overlap between contours that recurred in both sessions appear yellow. Lower left: target position of GRIN lens over the CA1 layer. Lower right: fluorescence image of lens position from the example rat. (**B**) Top: rastergram shows normalized calcium fluorescence traces of place cells (one per row, sorted by preferred firing location and direction) during several traversals of the short path during an example session. Bottom: rat's position (black) with running epochs colored by direction of travel ('LR,' left to right, in blue; 'RL,' right to left, in red). (**C**) Individual rat data (lines and symbols) and session means (bars) for the number of subsampled beeline trials (left) and median beeline running speed (right) during pre-training (*pre*) and training (*trn*) sessions; subsampled beeline trial counts were lower for *trn* than *pre* sessions because only trials from the first 10 min of *trn* (prior to shock delivery) were included; running speeds were lower for *trn* than *pre* sessions because scopolamine (SCP) reduced running speeds, resulting in preferential subsampling of slower beeline trials from drug-free (DF) sessions by the algorithm that minimized running speed differences between training conditions (see 'Methods'). (**D**) Bar/line graphs show number of place cells imaged per rat (left) and percentage of all imaged cells classified as place cells (right) during *pre* and *trn* sessions. (**E**) Pie graphs show percentages of place cells imaged per condition ('n' gives total number summed over rats) that were spatially tuned in the LR only, RL only, or both LR and RL running directions. (**F**) Tuning curve properties of place cells imaged during *pre* and *trn* sessions. (**G**) Place cell recurrence ratios (RR) between *pre* and *trn* sessions. (**H**) Top: diagram shows timeline for *pre* and *trn* sessions given 48 hr apart. Bottom: tuning curve heatmaps for recurring place cells (from all rats combined, co-sorted by peak locations from the *trn1* session) that were spatially tuned in the LR (top) or RL (bottom) running directions; separate heatmaps are shown for *pre*, *trn* part 1 (*trn1*), and *trn* part 2 (*trn2*) sessions. (**I**) Between- (B) and within- (W) session population vector correlation matrices are shown for DF and SCP shock training conditions; middle bar graph shows median place tuning stability (S) for each rat (lines and symbols) and mean over rats (bars) for B and W heatmap pairs. Asterisks in (**C**) and (**D**) denote significance for main effect of DF vs. SCP or uncorrected *t*-test comparing *pre* vs. *trn* sessions; asterisks in (**E**) and (**G**) denote significance for uncorrected *t*-tests. †p<0.1; *p<0.05; **p<0.01; ***p<0.001.

The online version of this article includes the following figure supplement(s) for figure 2:

**Figure supplement 1.** GRIN lens placements and cell contours for imaged neurons in CA1.

**Figure supplement 2.** Beeline trial counts and running speeds prior to downsampling.

**Figure supplement 3.** Relationship between place tuning versus size and eccentricity of imaged cell contours.

**Figure supplement 4.** Distributions of place cell tuning properties.

**Figure supplement 5.** Pre-training and training session data from individual rats.

**Figure supplement 6.** Between- and within-session place coding in male versus female rats.

*Figure 2 continued on next page*

Figure 2 continued

**Figure supplement 7.** Shuffled analysis of between- and within-session population coding.

**Figure supplement 8.** Pre-training and training session results from drug-free (DF) rats that failed to retain avoidance and scopolamine (SCP) rats that successfully retained avoidance.

7) that failed to avoid the short path 48 hr aftershock training (indicating that the drug had blocked avoidance learning). A 2 × 2 ANOVA on the number of place cells detected per session (*Figure 2D*, left) found no effect of training condition ($F_{1,14}$ = 0.03, p=0.87) or session ($F_{1,14}$ = 0.014, p=0.92), and no interaction ($F_{1,14}$ = 2.1, p=0.17). A similar ANOVA on the percentage of imaged cells per session that were classified as place cells (*Figure 2D*, right) found no effect of training condition ($F_{1,14}$ = 0.07, p=0.79), but place cell percentages were lower during training than pre-training sessions ($F_{1,14}$ = 5.1, p=0.04); this reduction was greater in the scopolamine than drug-free condition, resulting in a marginal interaction effect ($F_{1,14}$ = 4.4, p=0.0539). An analysis of how spatial tuning varied with the size and location of cell contours revealed that contours of non-place cells tended to be slightly but significantly smaller (p<0.01) and nearer to the lens center (p<0.05) than contours of place cells (*Figure 2—figure supplement 3A*).

For tuning curves that met the criteria for spatial selectivity in each session, four tuning properties were analyzed: peak Af/s rate, out-of-field Af/s rate, tuning width, and spatial information content (see 'Methods'). For each rat and each session, median values of these tuning properties were taken over all LR and RL tuning curves that met spatial selectivity criteria (*Figure 2—figure supplement 4*); 2 × 2 mixed ANOVAs were then performed on the resulting rate medians for each of the four tuning properties (*Figure 2F*). There were large main effects of session (pre-training vs. training) for all four tuning properties, which were driven by correspondingly large differences in behavior sampling and running speed (*Figure 2C*). However, there were no main effects of training condition (drug-free vs. scopolamine) or interactions between training condition and session (*Figure 2F*), indicating that none of the four tuning properties differed during training sessions given drug-free versus on scopolamine.

The *place field recurrence ratio* was defined as $RR = N_R/N_T$ , where $N_R$ is the number of place fields that recurred during the pre-training and training sessions, and $N_T$ is the total number of recurring and non-recurring place fields detected during both sessions combined (see 'Methods'). Mean $RR$ values were lower (independent $t_{14}$ = 2.15, p=0.0497) during scopolamine (40.7 ± 8.8%) than drug-free (50.7 ± 3.9%) training sessions, consistent with the findings reported above that the proportion of neurons classified as place cells during the training session was lower on scopolamine than drug-free (*Figure 2D*, right). It was observed that contours of recurring place cells tended to be slightly but significantly smaller (p<0.01) and nearer to the lens center (p<0.01) than contours of non-recurring place cells (see *Figure 2—figure supplement 3B and C*).

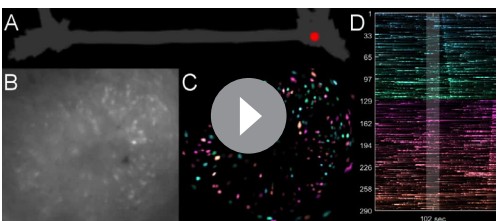

**Video 1.** Video demonstrating behavior and calcium imaging recorded from one rat performing linear alternation along the short path of the maze. (A) Position of the rat (red dot). (B) Motion-corrected video of calcium fluorescence. (C) Processed demixed/denoised CaImAn output (spatial contours modulated by denoised calcium trace). (D) Right half shows calcium activity of 290 identified neurons in the video, sorted by their preferred firing direction (teal/pink) and location along the track. Current time highlighted in the center gray region.

https://elifesciences.org/articles/80661/figures#video1

To investigate how scopolamine affected population coding by recurring place cells, two heatmaps (one per running direction, denoted $H_{LR}^{pre}$ and $H_{RL}^{pre}$) were created from pre-training tuning curves of recurring place cells (*Figure 2H*, *Figure 2—figure supplement 5*). Additionally, two heatmaps per running direction were created from data collected during the first ($H_{LR}^{trn1}$, $H_{RL}^{trn1}$) and second ($H_{LR}^{trn2}$, $H_{RL}^{trn2}$) half of each training session's subsampled beeline trials, all of which occurred prior to shock delivery. Between-session population vector correlation matrices (one per running direction) were derived for each rat (*Figure 2—figure supplement 5*) with matrix entries given by $\rho_{i,j}^{btwn} = R\left(H_i^{pre}, H_j^{trn1}\right)$, where $H_i^{pre}$ and $H_j^{trn1}$ denote population vectors in columns $i$ and $j$ of the pre-training and trn1 heatmaps, respectively, and $R$ is the Pearson correlation coefficient. Within-session population vector correlation

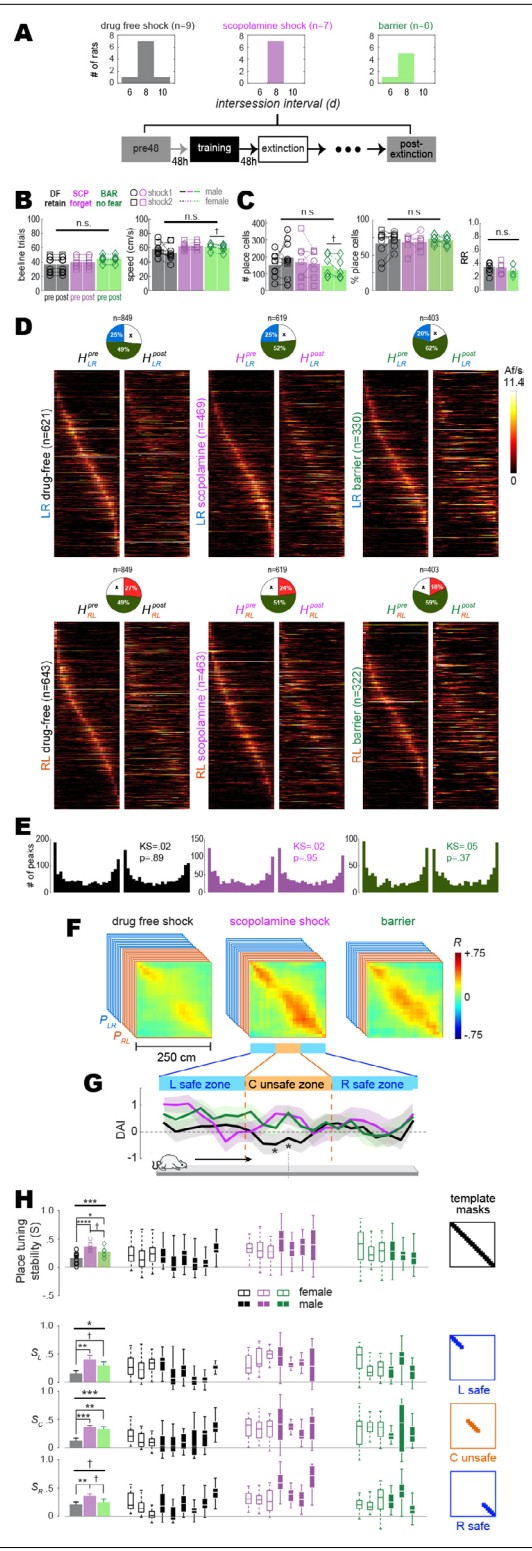

**Figure 3.** Remapping of place cells is induced by avoidance learning. (**A**) Top: bar graphs show distributions of intersession intervals between pre-training (pre) and post-extinction (post) session pairs included in the analysis. Bottom: diagram shows timeline for pre and post sessions. (**B**) Left: number of subsampled beeline trials did not differ significantly by training condition or session. Right: running speeds did not differ significantly by training condition; there was a marginal difference between pre vs. post running speeds in the barrier (but not drug-free or scopolamine shock) condition. (**C**) Left: total number of imaged place cells (recurring and non-recurring) did not

*Figure 3 continued on next page*

*Figure 3 continued*

differ significantly by training condition; there was a marginal difference between pre vs. post place cell counts in the barrier (but not drug-free or scopolamine shock) condition. Middle: percentage of all imaged cells (recurring and non-recurring) per session that were classified as place cells. Right: between-session place cell recurrence ratios did not differ by training condition. (**D**) Heatmap pairs show pre and post tuning curves for recurring place cells that were spatially tuned in the LR (top row) and RL (bottom row) running direction; both heatmaps in each pair are co-sorted by peak locations from the pre session. Pie charts show the proportion of recurring place cells (total number given at top) for which tuning curves were included in LR and RL heatmap pairs. (**E**) Kolmogorov–Smirnov tests show that distributions of place field peaks locations (LR and RL combined) were unchanged between pre and post sessions. (**F**) Mean population vector correlation matrices for the three training conditions. (**G**) Decoding accuracy index (DAI) at each position on the track for each of the three training conditions; '*' indicates locations where decoding was significantly (p<0.05) less accurate for the drug-free condition than the other two conditions. (**H**) Analysis of place tuning stability scores (S) along the full short path are shown in the top row; results for L, C, and R track zones are shown in the bottom 3 rows, respectively. Left column: bar graphs show means and standard errors of S; Middle columns: boxplots show median and range of template-selected population vector correlation bins in each rat. Right column: templates used to select peri-diagonal correlation values for analysis from different track zones. †p<0.1; *p<0.05; **p<0.01; ***p<0.001.

The online version of this article includes the following figure supplement(s) for figure 3:

**Figure supplement 1.** Pre-training and post-extinction session data from individual rats.

**Figure supplement 2.** Remapping in male versus female rats.

**Figure supplement 3.** Pre-training and post-extinction results from drug-free (DF) rats that failed to retain avoidance and scopolamine (SCP) rats that successfully retained avoidance.

**Figure supplement 4.** Place field shifts induced by remapping.

**Figure supplement 5.** Shuffled analysis of remapping.

---

matrices were derived in a similar manner with matrix entries given by $\rho_{i,j}^{wthn} = R\left(H_i^{trn1}, H_j^{trn2}\right)$, where $H_i^{trn1}$ and $H_j^{trn2}$ denote population vectors in columns $i$ and $j$ of the trn1 and trn2 heatmaps, respectively. When population vector correlation matrices were averaged together across running directions and rats (*Figure 2I*), high mean $\rho^{wthn}$ values were observed along diagonals of within-session correlation matrices for both the drug-free and scopolamine conditions. By contrast, lower mean $\rho^{btwn}$ values were observed along diagonals of between-session correlation matrices, in accordance with prior studies showing that place cell population vectors become decorrelated over time (*Mankin et al., 2012*; *Ziv et al., 2013*; *Cai et al., 2016*; *Keinath et al., 2022*). To quantify this decorrelation, a place tuning stability score (S) was computed as $S = median(\rho_1^{LR} - \rho_{off}^{LR}, \rho_2^{LR} - \rho_{off}^{LR}, ..., \rho_K^{LR} - \rho_{off}^{LR}, \rho_1^{RL} - \rho_{off}^{RL}, \rho_2^{RL} - \rho_{off}^{RL}, ..., \rho_K^{RL} - \rho_{off}^{RL})$, where $\rho_{off}^{LR}$ and $\rho_{off}^{RL}$ are mean off-diagonal $\rho$ values and $\rho_i^{LR}$ and $\rho_i^{RL}$ are individual peri-diagonal $\rho$ values for bins in the rat's LR and RL correlation matrices (see 'Methods'). A 2 × 2 mixed ANOVA on S scores with training condition (drug-free vs. scopolamine) as an independent factor and time interval (between vs. within session) as a repeated factor found no main effect of training condition ($F_{1,14} = 1.73$, p=0.21), but there was a large main effect of time interval ($F_{1,14} = 71.5$, p=7.1e⁻⁷) and a significant interaction ($F_{1,14} = 8.1$, p=0.0132). Uncorrected post hoc comparisons confirmed that this was because scopolamine injections reduced between- ($t_{14} = 3.1$, $S_{DF} < S_{SCP}$ p=0.0084) but not within- ($t_{14} = 0.83$, $S_{DF} < S_{SCP}$ p=0.42) session population vector correlations (*Figure 2I*, center). Hence, even though scopolamine did not acutely disrupt spatial tuning of CA1 neurons (*Figure 2F*), the drug reduced the percentage of recurring place cells and degraded the between-session (but not within-session) similarity of recurring place cell population vectors (*Figure 2H and I*). Similar disruption of between- but not within-session place cell stability has been previously reported in aged rats (*Barnes et al., 1997*).

Disruption of between-session place cell stability by scopolamine was similar in male and female rats (*Figure 2—figure supplement 6*). Cross-session stability of population vectors was completely abolished in control analyses (see 'Methods') where inferred spikes were circularly shifted against position tracking data during each individual beeline trial of the pre-training reference session (*Figure 2—figure supplement 7*). When analyses of *Figure 2* were repeated on scopolamine rats (n = 2) that retained avoidance despite the injection, population vector similarity was not degraded between pre-training and training sessions, suggesting that impairment of between-session place tuning stability

was related to impairment of avoidance learning (*Figure 2—figure supplement 8*). When the analyses were repeated on drug-free rats (n = 2) that failed to retain avoidance despite being trained drug free, between- and within-session population vector similarity were indistinguishable from drug rats that retained avoidance (*Figure 2—figure supplement 8*), implying that failure to acquire avoidance in these rats was not attributable to any noticeable deficiency in the fidelity of the hippocampal place code during training.

## Induction of place cell remapping by aversive learning

Training-induced place cell remapping was analyzed in the same subset of rats shown in *Figure 2*. To measure remapping, post-extinction CA1 population vectors were compared against the 48 hr pre-acquisition baseline session. For the drug-free shock condition, post-extinction population vectors were sampled from the first session aftershock training during which the rat met criterion (>2 rewards/min, and twice as many short paths as long paths) for extinction of short path avoidance (*Figure 3A*, left); this was the earliest post-training session with adequate sampling of the short path. For the scopolamine shock condition, post-extinction population vectors were sampled from the session given 8 d (i.e., four sessions including the training session) after the pre-training session (*Figure 3A*, center); this matched the modal intersession interval for the drug-free condition, so that confounds would not arise from measuring remapping across different time intervals for the drug-free versus scopolamine conditions. Likewise, for the barrier control condition, post-extinction population vectors were also sampled 8 d after the pre-training session; one barrier training rat became startled due to external factors and stopped running early in the 8 d post-training session, so the 6 d session was used instead (*Figure 3A*, right).

To control for potential confounds from between-session behavior differences, beeline trials were subsampled using an algorithm (see 'Methods') that equalized the number of trials from pre- and post-extinction sessions in each rat (*Figure 3B*, left), while also minimizing running speed differences between sessions and training conditions (*Figure 3B*). Spatial tuning was analyzed during subsampled trials, and a 3 × 2 mixed ANOVA on the number of place cells detected during each session (*Figure 3C*, left) yielded no effect of training condition ($F_{2,19}$ = 0.43, p=0.66) or session ($F_{1,19}$ = 0.01, p=0.92), and no interaction effect ($F_{2,19}$ = 1.57, p=0.24). A similar ANOVA on the percentage of imaged cells per session classified as place cells also yielded no significant effects (training condition: $F_{2,19}$ = 0.14, p=0.87; session: $F_{1,19}$ = 0.56, p=0.46; interaction: $F_{2,19}$ = 1.28, p=0.3). The place field recurrence ratio (*RR*) between pre- and post-extinction sessions also did not differ among training conditions (*Figure 3C*, right; one-way ANOVA: $F_{2,19}$ = .65, p=0.53).

To investigate whether shock-induced remapping occurred, a pair of heatmaps (pre and post) was created from tuning curves of recurring place cells in each running direction (LR and RL) for each training condition (*Figure 3D*). As in prior calcium imaging studies (*Ziv et al., 2013*), place fields tended to be more densely concentrated near the ends of the short path than in the middle (*Figure 3E*). Heatmap pairs were used to derive two population vector correlation matrices per rat, $P_{LR}$ and $P_{RL}$ (*Figure 3—figure supplement 1*). When $P_{LR}$ and $P_{RL}$ were averaged across rats, a gap of low correlation appeared in the center of the diagonal for the drug-free shock condition (*Figure 3F*, left), which was not seen in the scopolamine shock (*Figure 3F*, middle) or barrier (*Figure 3F*, right) conditions, suggesting that drug-free shock training induced more remapping of the short path than scopolamine shock or barrier training, in accordance with prior studies (*Moita et al., 2003*; *Moita et al., 2004*; *Wang et al., 2012*; *Mamad et al., 2019*; *Schuette et al., 2020*; *Ormond et al., 2023*).

To statistically analyze this remapping effect, a place tuning stability score *S* was derived for each rat by the methods described above. Boxplots in *Figure 3H* (top row) show each rat's median and range of *S* values from all peridiagonal bins in $P_{LR}$ and $P_{RL}$ combined. Bar graphs (*Figure 3H*, top left) for mean stability scores, $\overline{S}$, for each training condition differed significantly between drug-free, scopolamine, and barrier conditions (one-way ANOVA: $F_{2,19}$ = 9.7, p=0.0012); uncorrected post hoc *t*-tests confirmed that this was because $\overline{S}$ was lower in the drug-free shock than scopolamine shock ($t_{14}$ = 4.3, p=0.0007) or barrier ($t_{13}$ = 2.2, p=0.0498) conditions, which differed only marginally from one another ($t_{11}$ = 1.98, p=0.0834). Hence, drug-free avoidance training and extinction induced more place cell remapping than the scopolamine shock or barrier training conditions; this remapping occurred at all locations on the short path but a 3 × 3 repeated-measures ANOVA with training condition (drug-free shock, scopolamine shock, barrier) and track zone (left safe, center shock, right safe; bottom 3

rows of *Figure 3H*) as factors suggested that remapping was most pronounced in the center zone where shocks occurred than at L and R ends of the track where no shocks occurred (*Figure 3H*, bottom; $F_{2,40} = 3.0$, p=0.0596). Induction of remapping by avoidance training and extinction did not differ in male versus female rats, nor did the impairment of remapping by scopolamine (*Figure 3—figure supplement 2*). When the analyses of *Figure 3* were repeated on scopolamine rats (n = 2) that successfully retained avoidance despite the injection, the drug was less effective at inducing place cell remapping (*Figure 3—figure supplement 3*), suggesting (but not proving) that impairment of remapping was causally related to impairment of avoidance learning. When the analyses were repeated on drug-free rats (n = 2) that failed to retain avoidance despite being trained drug free (*Figure 3—figure supplement 2*), one rat showed less and the other showed more place tuning stability across the entire length of the track when compared against rats that retained avoidance.

Kolmogorov–Smirnov tests found that place field density distributions (accumulated over all rats) did not change significantly between the pre versus post sessions in any training condition, including the drug-free shock condition (*Figure 3E*, left). Hence, remapping did not cause place fields to migrate toward or away from the shock zone (*Figure 3—figure supplement 4*), in contrast with prior reports that remapping can cause place fields to migrate toward locations where aversive stimuli are encountered (*Mamad et al., 2019*). All place cell remapping effects and differences among training conditions were abolished when remapping analyses were performed on shuffled data (*Figure 3—figure supplement 5*).

To analyze the extent of remapping caused by aversive learning, the rat's position on the track was decoded from place cell calcium activity (see 'Methods'). The decoder was trained on data from one session (pre- or post-extinction) and tested on data from the other session. The decoding error in a given position bin was measured as the median of the absolute difference (in cm) between the rat's actual versus predicted position over all visits to that bin. Wilcoxon signed-rank tests compared decoding errors obtained with shuffled versus non-shuffled population vectors; a *decoding accuracy index* (DAI) was then derived for each bin, with DAI = 0 corresponding to p=0.01 for more accurate decoding from unshuffled than shuffled population vectors (see 'Methods'). When mean DAI values were compared among training conditions, significant differences were detected only in two bins at the center of the track (in the shock zone) where decoding in the drug-free shock condition was significantly less accurate (p<0.05) than in scopolamine shock and barrier training conditions combined (asterisks in *Figure 3D*, bottom). In accordance with results from population vector correlation analysis, these decoding results support the conclusion that drug-free shocks induced more remapping of the center zone than shocks while on scopolamine or from barrier training.

## Shock-evoked responses of place cells

Shock-induced place cell remapping has been observed in prior studies (*Moita et al., 2003*; *Moita et al., 2004*; *Wang et al., 2012*; *Schuette et al., 2020*), but the cellular and synaptic mechanisms of such remapping are not well understood. Individual CA1 neurons can form new place fields at locations where they are artificially stimulated or inhibited (*Bittner et al., 2015*; *Geiller et al., 2022*), and since some CA1 place cells are naturally responsive to shocks (*Moita et al., 2003*; *Moita et al., 2004*), it is possible that individual CA1 neurons might gain or lose place fields at locations where they respond to shocks. If so, then this might partly account for remapping at the population level. If shock-evoked responses of CA1 neurons are involved in remapping, then blockade of remapping by scopolamine could be related to effects of the drug upon these shock-evoked responses. To investigate this, each place cell imaged during a training session was classified as shock-responsive if its Af/s rate was significantly higher (p<0.01) during entries to the electrified than non-electrified shock zone; otherwise, it was classified as non-responsive to shocks (see 'Methods).

The mean percentage of shock-responsive cells per rat (*Figure 4A*, left column) was 35.5 ± 2.3% for drug-free training and 29.6 ± 3.3% for scopolamine training; these percentages did not differ significantly (independent $t_{14} = 1.58$; p=0.13). Pre-shock baseline Af/s rates (*Figure 4A*, middle column) were higher for shock-responsive than non-responsive place cells in both the drug-free ($t_{1931} = 9.3$; p<0.0001) and scopolamine ($t_{1034} = 4.34$; p<0.0001) conditions, which was attributable to the fact that in both conditions, place field peaks for shock-responsive and non-responsive cells tended to be segregated on the 'approach' and 'departure' sides of the shock zone, respectively (*Figure 4A*, right column). Thus, as rats approached the shock zone in either running direction, shock-responsive cells

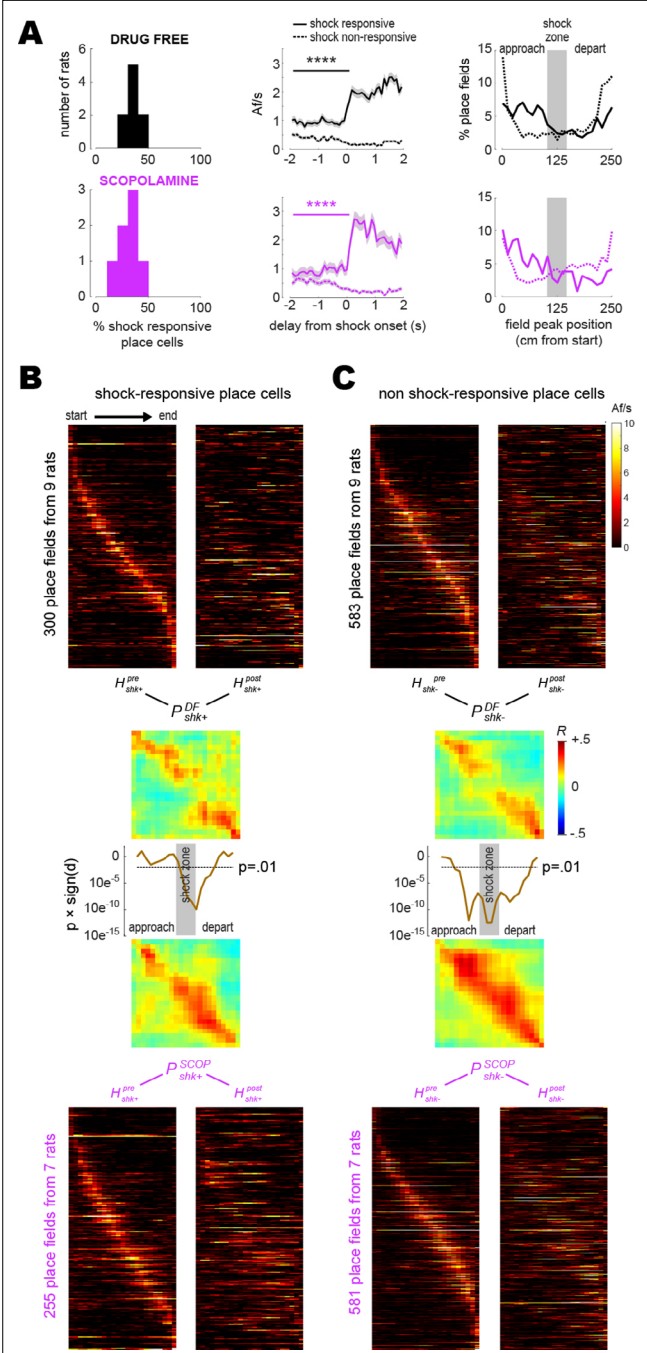

**Figure 4.** Shock-evoked responses of place cells. (**A**) Left column: frequency distribution of shock-responsive place cell percentages in each rat. Middle column: population-averaged Af/s rates (from all rats combined) for shock-responsive versus non shock-responsive place cells. Right column: spatial distribution of place field peak locations during the training session, prior to grid electrification; x-axis shows distance traveled between the start (left) and end (right) of the short path in the current running direction (LR or RL). (**B**) Heatmaps (LR and RL tuning curves combined) are plotted for shock-responsive place cells in the drug-free (top) versus scopolamine (bottom) conditions during pre- (left) and post- (right) training sessions (rows are co-sorted on peaks from the pre-training session, columns are sorted so that the rat's running direction goes from left to right for both LR and RL tuning curves); P matrices show pre- vs. post-extinction population vector correlations for shocks given drug-free (top) or on scopolamine (bottom); line graphs show that signed p-values (y-axis, log scale) fall below .01 at center track positions where drug-free shocks induced more remapping than shocks on scopolamine. (**C**) Same as (**B**) but for shock non-responsive place cells. ****p<0.0001.

were more likely than non-responsive cells to be active in their place fields just before shock zone entry.

To compare remapping by shock-responsive versus non-responsive CA1 neurons, analyses were performed on place cells that recurred during pre- and post-extinction sessions and were also active during the training session (so that each cell could be classified as responsive or non-responsive to shocks). Pre- and post-extinction heatmaps were derived from tuning curves of shock-responsive (*Figure 4B*) and non-responsive (*Figure 4C*) place cells. LR and RL tuning curves were combined together in each heatmap, with RL (but not LR) tuning curves reflected on the horizontal so that the rat's running direction was oriented from left to right in all rows; hence, columns to the left or right of center in each heatmap denoted locations where rats were approaching or departing the shock zone, respectively. From these heatmaps, shock-responsive and non-responsive population vector correlation matrices were derived for the drug-free ($P_{shk+}^{DF}$, $P_{shk-}^{DF}$) and scopolamine ($P_{shk+}^{SCOP}$, $P_{shk-}^{SCOP}$) conditions. A gap of low population vector correlation values in the center of $P_{shk+}^{DF}$ but not $P_{shk+}^{SCOP}$ (*Figure 4B*) and of $P_{shk-}^{DF}$ but not $P_{shk-}^{SCOP}$ (*Figure 4C*) indicated that drug-free shocks induced remapping of the center zone in both shock-responsive and non-responsive place cell populations, whereas scopolamine shocks failed to induce remapping in either population. Confirming this, a series of independent *t*-tests was performed on clusters of peri-diagonal bins arrayed along diagonals of $P_{shk+}^{DF}$ versus $P_{shk+}^{SCOP}$, and $P_{shk-}^{DF}$ versus $P_{shk-}^{SCOP}$ (see 'Methods'); resulting significance values (line graphs in *Figure 4B and C*) dipped well below p<0.01 in the center region of the track (but not at the ends), indicating that drug-free shocks caused shock-responsive and non-responsive place cells to remap the center zone more than scopolamine shocks.

## Discussion

Place cell remapping has been shown to occur at locations where animals experience aversive events such as shocks, predator odors, or attacks (*Moita et al., 2003*; *Moita et al., 2004*; *Wang et al., 2012*; *Wang et al., 2015*; *Kim et al., 2015*; *Mamad et al., 2019*; *Schuette et al., 2020*; *Ormond et al., 2023*), but it is not well understood whether such remapping plays a functional role in storing memories for aversive encounters. Here we investigated this question by using a large-field-of-view version of the UCLA Miniscope (*Guo et al., 2023*) to image population activity of CA1 place cells in rats as they acquired and extinguished a spatial avoidance learning task. We found that significantly more place cell remapping was induced under experimental conditions where aversive encounters were remembered (drug-free shocks) than under control conditions where they were forgotten (shock training on scopolamine), or where a neutral rather than aversive path-blocking stimulus was encountered (barrier training). While these results do not definitively prove that place cell remapping is causally necessary for storing memories of aversive encounters, they provide correlational evidence that remapping occurs selectively under conditions where a stimulus occurs and is subsequently remembered rather than forgotten.

### Spatial distribution of place fields

Hippocampal place fields are sometimes observed to be preferentially concentrated at locations where rodents spend large amounts of time, such as the two rewarded locations on our rectangular maze (*Figure 3E*). This makes sense from an information theoretic perspective since it is efficient for frequently occurring stimuli to be overrepresented (and conversely for rarely occurring stimuli to be underrepresented) among preferred stimuli of tuned neurons in a population code. In accordance with this principle, when animals learn to obtain reward at specific locations (and thus to spend more time at those locations), a form of remapping occurs whereby place fields migrate toward rewarded locations (*Breese et al., 1989*; *Hollup et al., 2001*; *Dupret et al., 2010*; *Mamad et al., 2017*).

Less is known about how aversive learning affects the spatial distribution of place fields, but based purely upon information theoretic considerations, one might expect that place fields should migrate away from (rather than toward) aversively reinforced locations as animals shift their behavior to spend less time there. Contradicting this intuition, there is evidence to suggest that place fields migrate toward (rather than away from) aversively reinforced locations (*Mamad et al., 2019*). Possibly, overrepresentation of aversively reinforced locations increases the likelihood that place cells encoding such locations will become active during 'replay events' that occur during sleep (*Girardeau et al.,*

*2017*), or during waking as the animal deliberates over whether to approach a remote location where danger has been encountered (*Wu et al., 2017*; *Ormond et al., 2023*). This may help animals to avoid dangerous places by anticipating in advance the threats that await them there.

We analyzed place cell remapping after rats first learned to avoid a shocked location, then extinguished avoidance and resumed spending time at the formerly shocked location. In agreement with prior findings (*Moita et al., 2003*; *Moita et al., 2004*; *Wang et al., 2012*; *Schuette et al., 2020*), we observed that place cell population vectors were less correlated (and thus showed more remapping) in rats that underwent acquisition and extinction of avoidance than in control rats that encountered a neutral barrier or failed to retain avoidance after being trained under the influence of scopolamine (*Figure 3F–H*). In contrast with *Mamad et al., 2019*, place cells did not shift their center of mass toward the site of the aversive stimulus; instead, the spatial distribution of place fields remained unchanged by remapping (*Figure 3E*), indicating that place fields shifted toward and away from the shocked location in equal proportions.

The discrepancy in place field migration findings might be related to our use of calcium imaging rather than electrophysiology to record neural activity; mChARs can regulate calcium entry into cells via a wide range of mechanisms (*Brown, 2018*) and thus scopolamine may have different effects on spike detection from calcium traces versus electrophysiological recordings. The discrepancy might also be a consequence of using different aversive stimuli (shock versus noxious odor), different degrees of fear acquisition and extinction during the task, or differences in the spatial layout of the maze. Rats in our experiment only received one or two self-initiated shocks during the shock sessions, thus a stronger conditioning paradigm might be necessary to induce place field migration. Another possible source of discrepancy could be different methods for normalizing speed and occupancy across experimental conditions. Aversive events cause lasting behavioral changes that alter the manner in which an animal samples its spatial environment (including reduced movement speed and avoidance of locations where aversive stimuli have been encountered). To control for confounding effects of these learning-induced behavior changes upon measures of place cell remapping, studies rely upon subsampling methods to normalize speed and occupancy distributions from experimental sessions conducted before versus after encounters with aversive events (*Moita et al., 2003*; *Moita et al., 2004*; *Wang et al., 2012*; *Schuette et al., 2020*). Here, we minimized confounds of behavioral sampling by restricting data analysis to 'beeline trials' during which rats ran highly stereotyped trajectories on a linear track, so that behavior was matched as closely as possible between sessions conducted before acquisition versus after extinction of avoidance learning. Using this approach, we replicated prior findings that aversive learning induces place cell remapping (*Moita et al., 2003*; *Moita et al., 2004*; *Wang et al., 2012*; *Wang et al., 2015*; *Kim et al., 2015*; *Mamad et al., 2019*; *Schuette et al., 2020*; *Ormond et al., 2023*) but found no change in the spatial distribution of place fields.

## Acute effects of scopolamine

Significantly more place cell remapping occurred when aversive encounters were remembered (as assessed by 48 hr avoidance expression) after drug-free training than when they were forgotten after training on scopolamine. This suggests that place cell remapping might play a functional role in storing memories of aversive encounters, but potential confounds must be considered. A trivial explanation for our finding that scopolamine impaired both avoidance learning and place cell remapping could be that, at the dose given, the drug impaired rats' awareness of their surroundings and their ability to perceive shock stimulus during the training session. This interpretation is not well supported by the data for several reasons. First, scopolamine did not acutely affect median running speeds during beeline trials (*Figure 2C*), consistent with previous demonstrations that 1 mg/kg scopolamine does not impair locomotion in rodents (*Sun et al., 2021*; *Anagnostaras et al., 1999*). Second, scopolamine did not acutely impair immediate avoidance of the shocked location during training (*Figure 1F*), consistent with prior evidence that mAChRs are not necessary for immediate shock avoidance (*Decker et al., 1990*). Third, under the influence of scopolamine, place cells retained stable within-session spatial tuning (*Figure 2F–I*) and exhibited shock-evoked responses similar to those observed in the drug-free condition (*Figure 4A*). By contrast, a recent calcium imaging study in mice (*Sun et al., 2021*) reported that tuning properties of place cells on a linear track were acutely disrupted by systemic scopolamine injections at the same dosage used here (1 mg/kg) (but see also *Brazhnik et al., 2004*). More consistent with our current findings, electrophysiology studies in rats have reported that

systemic ACh antagonists alter temporal firing properties of place cells, such as their synchronization with hippocampal theta rhythm, while largely preserving their spatial firing properties (*Douchamps et al., 2013*; *Newman et al., 2017*). Conversely, cholinergic activation of the hippocampus acutely alters EEG states by promoting theta oscillations and suppressing sharp-wave ripples (*Zylla et al., 2013*; *Vandecasteele et al., 2014*; *Hunt et al., 2018*; *Ma et al., 2020*; *Jarzebowski et al., 2021*). Therefore, it is possible that theta oscillations and sharp waves were acutely altered by scopolamine during our experiments, but these signals were not detectable using our calcium imaging methods, so it was not possible to analyze these phenomena in the current study.

Although rats in our study were still able to locomote through the environment and keep track of their own location while on scopolamine, some acute effects of the drug upon behavior and place cell activity were observed. For example, while on scopolamine, rats showed a significant reduction of their previously learned preference for the short path (*Figure 1F*) prior to any delivery of shocks. This suggests that scopolamine may have affected the animals' motivation to seek reward on the maze or impaired recall of previously learned strategies for finding the reward, consistent with prior evidence that scopolamine can impair navigational learning and decision-making (*Nieto-Escámez et al., 2002*; *Vales and Stuchlik, 2005*; *Huang et al., 2011*; *Svoboda et al., 2017*; *Rashid and Ahmed, 2019*; *Hales et al., 2020*). Although place cells continued to exhibit spatial tuning while rats were on scopolamine, between-session place cell recurrence probabilities were reduced (*Figure 2G*) and population vectors showed degraded between-session similarity with drug-free maze visits on prior days (*Figure 2H and I*); interestingly similar impairment of between-session place cell stability has been reported in aged rats (*Barnes et al., 1997*).

By selectively degrading between-session (but not within-session) stability of place cells, scopolamine may have prevented CA1 from retrieving accurate memory representations of the familiar maze environment during the training session. If so, then this could help to explain why rats acutely stopped preferring the short path while on scopolamine; expressing the learned preference for the short path may depend upon recalling stored maze representations that were previously active when the preference was learned. It may also help to explain why scopolamine prevented the aversive shock experience from inducing long-term remapping of place cells since failure to recall previously stored maze representations during training may have shielded those representations from modification by the aversive experience, and thus from remapping.

## Effects of scopolamine on memory storage and retrieval

Many studies have shown that ACh receptor antagonist drugs can impair hippocampal memory processing in animal models (*Decker et al., 1990*; *Anagnostaras et al., 1999*; *Atri et al., 2004*; *Hasselmo and McGaughy, 2004*; *Green et al., 2005*; *Hasselmo, 2006*; *Huang et al., 2011*; *Dannenberg et al., 2017*; *Solari and Hangya, 2018*). Conversely, drugs that upregulate ACh neurotransmission can facilitate learning and memory (*Barten and Albright, 2008*; *Digby et al., 2010*; *Ragozzino et al., 2012*) and are commonly prescribed to treat aging-related amnesia and dementia in human patients (*Lombardo and Maskos, 2015*). Here, rats that received avoidance training on scopolamine were impaired when tested for retention 48 hr later (*Figure 1H*), consistent with prior studies showing that mAChRs are necessary for acquisition of aversive learning (*Decker et al., 1990*; *Anagnostaras et al., 1995*; *Anagnostaras et al., 1999*; *Wallenstein and Vago, 2001*). In conjunction with blocking avoidance learning, scopolamine also blocked place cell remapping between pre-training and post-extinction sessions (*Figure 3D–H*). Prior studies have similarly shown that scopolamine interferes with the ability of place cells to form and update cognitive maps across repeated visits to the same environment (*Douchamps et al., 2013*), but to our knowledge, the present study is the first to show that impairment of memory for an episodic event (in this case, an aversive shock) by scopolamine is accompanied by disruption of place cell remapping that normally occurs in response to such an event.

Since rats both acquired and extinguished avoidance behavior during the timespan over which remapping was measured, it was not possible to fully dissociate whether place cell remapping in drug-free rats was driven mainly by acquisition, extinction, or both. Rats trained on scopolamine did not undergo extinction (since they failed to exhibit long-term retention of avoidance behavior in the first place), so if remapping in drug-free rats was driven mainly by extinction, then scopolamine may have impaired remapping not by disrupting learning during the training session but instead by depriving rats of the extinction experience that would otherwise have caused remapping to occur

after training. Prior evidence indicates that acquisition and extinction of conditioned fear can both contribute to place cell remapping (*Moita et al., 2004*; *Wang et al., 2012*; *Wang et al., 2015*), so it is likely that impairment of remapping by pre-training injections of scopolamine was at least partly caused by disruption of experience-dependent changes to the hippocampal population code during the training session.

Manipulations that impair ACh neurotransmission have been found to selectively impair storage but not retrieval in certain memory tasks, such as remembering semantic information in humans (*Atri et al., 2004*; *Hasselmo and McGaughy, 2004*) and fear learning in rodents (*Decker et al., 1990*; *Huang et al., 2011*). Hippocampal synaptic plasticity is likewise thought to be more important for storage than retrieval of hippocampal-dependent memory tasks (*Martin et al., 2000*). Hippocampal plasticity is sensitive to mAChR antagonists (*Drever et al., 2011*) and has been implicated in place cell remapping (*Dragoi et al., 2003*), so it is possible that in our study scopolamine impaired avoidance learning and place cell remapping by interfering with synaptic plasticity in the hippocampus, and thus preventing the storage of memories for the aversive shock experience. Some CA1 place cells are excited by shocks (*Moita et al., 2003*; *Moita et al., 2004*), and it has been shown that individual CA1 neurons can form new place fields at locations where they are artificially excited or inhibited (*Bittner et al., 2015*; *Schoenenberger et al., 2016*; *Geiller et al., 2022*; *McKenzie et al., 2021*). This raises the possibility that during acquisition of fear learning, shock-evoked responses of CA1 neurons might trigger synaptic plasticity that leads to place cell remapping. If scopolamine disrupted shock-evoked hippocampal plasticity, it likely did so by impairing mechanisms downstream from neural responses to shock since we found that scopolamine had no effect on shock-evoked responses of CA1 place cells and shock-induced remapping was similarly prevalent in subpopulations of shock-responsive and non-responsive place cells.

Since the drug was given systemically, scopolamine may have interfered with plasticity not only in the hippocampus but also in other brain regions that contribute to the storage of fear memories, such as the amygdala, where associations between predictive cues and aversive events are thought to be stored by synaptic plasticity that is both triggered by noxious stimuli (*Blair et al., 2001*; *Grewe et al., 2017*; *d'Aquin et al., 2022*) and susceptible to cholinergic modulation (*Izquierdo et al., 1992*; *Jiang et al., 2016*). Regardless of whether scopolamine disrupted place cell remapping via its effects upon synaptic plasticity or other mechanisms (such as retrieval of previously stored maze respresentations; see above), or whether the drug acted upon pharmacological targets inside versus outside of the hippocampus (or both), our findings indicate that the drug disrupted shock-induced place cell remapping and thus support the idea that reorganization of the hippocampal place code occurs in conjunction with hippocampal storage of episodic memories for aversive events. Where and how scopolamine acts in the brain to impair remapping is a worthwhile question for further research that could help to elucidate the broader role of acetylcholine in memory storage and retrieval.

## Implications for brain disease and aging

If remapping is a mechanism for memory, then disruption of remapping by systemic cholinergic antagonists may serve as a useful animal model for memory loss accompanied by degeneration of cholinergic signaling in dementia (*Wenk et al., 1980*; *Arendt et al., 1983*; *Hardy, 2006*; *Schliebs and Arendt, 2011*). Degradation of cholinergic signaling also occurs with alcohol abuse disorder (*Arendt, 1994*; *Miller and Kamens, 2020*). By further investigating how the cholinergic neuromodulatory system interacts with the hippocampus, we may develop a deeper understanding of how memory systems are impacted by a variety of neurodegenerative insults and make progress toward improved therapies for memory diseases.

Like humans, rodents show memory deficits with aging (*Barnes, 1979*; *Barnes et al., 1980*). Interestingly, prior studies have shown that place cell population dynamics in aged rats are altered in ways similar to those observed here under scopolamine. For example, we found that scopolamine degraded between- but not within-session stability of the CA1 population code, which has also been observed in aged rats (*Barnes et al., 1997*). This is in contrast to other brain diseases, such as epilepsy, which degrades both within- and between-session place coding (*Shuman et al., 2020*). Additionally, we found that in rats that showed impaired avoidance learning after pre-training injections of scopolamine, the drug concurrently prevented CA1 place cells from remapping during learning and extinction (*Figures 3 and 4*). Place cells in aged rodents have likewise been reported to show deficits

in experience-dependent modification of their tuning properties (*Oler and Markus, 2000*; *Wilson et al., 2003*; *Wilson et al., 2004*; *Tanila et al., 2018*). These parallels between the effects of aging and scopolamine upon place cells suggest that mAChR blockade and aging may produce similar memory deficits by causing similar impairments to hippocampal population stability. Further study is warranted to investigate whether such impairments of hippocampal population coding can be reversed using anti-amnestic drugs, such as those that upregulate ACh transmission or other novel compounds. By identifying interventions that can improve the long-term stability of hippocampal population coding in animal models, it may be possible to make progress toward improved therapies for memory deficits associated with aging and neurodegenerative disease.

## Methods

All experimental procedures were approved by the Chancellor's Animal Research Committee of the University of California, Los Angeles, in accordance with the US National Institutes of Health (NIH) guidelines, protocol #2017-038. Raw data files and Matlab analysis code are openly available at https://github.com/tadblair/tadblair or https://doi.org/10.5068/D1ZT2S.

### Subjects and apparatus

15 Long–Evans rats (eight females, seven males) acquired from Charles River at 3 months of age were used in this experiment. Subjects were singly housed within a temperature and humidity controlled vivarium on a 12 hr reverse light-cycle. Beginning at 4 months of age, rats were reduced 85% of their ad lib weight (450–550 g) through limited daily feeding, and then underwent virus injection and GRIN lens implantation surgery (see below). After 5 d recovery from surgery, rats began a regimen of behavior and recording sessions on the rectangular maze. The maze was constructed from a 4″ wide track composed from metal grid bars along the full length of the long and short paths. The track was elevated 20″ from the floor. There were extramaze cues in the surrounding environment (doors, shelving, checkerboard poster on the wall) but no local cues on the maze to indicate the shock zone's location. During shock training sessions (drug-free or scopolamine), a 1.0 mA current from a grid scrambler (Med Associates Corporation, Fairfax, VT) was switched on to electrify grid bars in a 50 cm segment of the short path's center (*Figure 1A*). The exact duration of the rat's exposure to shock varied with the length of time it spent in contact with the electrified bars as it ran across or retreated from the shock zone, but was typically very brief (on the order of 1 s or less).

*Figure 1B* shows the order in which experimental manipulations were given to different groups of rats. Six rats (three males, three females) received drug-free shock training as their first manipulation after reaching criterion for preferring the short path. Of these, five rats (three males, two females) then received extinction sessions and ended the experiment. The remaining female rat was mistakenly given shock at the start of the drug-free training session (rather than after 10 min) and was thus dropped from imaging analysis of the drug-free shock condition, but after extinction, this rat subsequently received scopolamine shock training and was thus included in analysis for that condition (see below). Four rats (two males, two females) were given their drug-free shock sessions after previously receiving barrier training as their first manipulation (so these rats were still naive to shock during drug-free shock avoidance training). Three rats (one male, two females) were given their drug-free shock session after previously receiving a scopolamine shock session (followed by extinction) as their first manipulation, and two rats (one male, one female) were given their drug-free shock session after previously receiving both barrier training and a scopolamine shock session (followed by extinction). Among rats that were given scopolamine shock sessions, five (two males, three females) had not previously experienced shock at the time of their scopolamine shock avoidance training session, whereas five rats (three males, two females) had previously received drug-free shock avoidance training followed by extinction. To compare imaging results across training conditions, rats from the drug-free condition were only included if they ran <60% as many trials during their first post-extinction session as during their final pre-training session (indicating that they had successfully acquired short path avoidance), whereas rats from the scopolamine shock condition were only included if they ran >60% as many trials during their first post-extinction session as during their final pre-training session (indicating that scopolamine had successfully blocked avoidance learning).

## Surgical procedures

Subjects were given two survival surgeries prior to behavior training in order to record fluorescent calcium activity from hippocampal CA1 cells. During the first surgery, rats were anesthetized with 5% isoflurane at 2.5 L/min of oxygen, then maintained at 2–2.5% isoflurane while a craniotomy was made above the dorsal hippocampus. Next, 1.2 uL of AAV9-Syn-GCamp7s (AddGene) was injected at 0.12 uL/min just below the dorsal CA1 pyramidal layer (–3.6 AP, 2.5 ML, 2.6 DV) via a 10 uL Nanofil syringe (World Precision Instruments) mounted in a Quintessential Stereotaxic Injector (Stoelting) controlled by a Motorized Lab Standard Stereotax (Harvard Apparatus). Left or right hemisphere was balanced across all animals. One week later, the rat was again anesthetized and four skull screws were implanted to provide stable hold for the subsequent implant. The viral craniotomy was reopened to a diameter of 1.8 mm, and cortical tissue and corpus callosal fibers above the hippocampus were aspirated away using a 27- and 30-gauge blunt needle. Following this aspiration, and assuring no bleeding persisted in the craniotomy, a 1.8-mm-diameter Gradient Refractive INdex lens ('GRIN lens,' Edmund Optics) was implanted over the hippocampus and cemented in place with methacrylate bone cement (Simplex-P, Stryker Orthopaedics). The dorsal surface of the skull and the bone screws were cemented with the GRIN lens to ensure stability of the implant, while the dorsal surface of the implanted lens was left exposed.

Two to three weeks later, rats were again placed under anesthesia in order to cement a 3D-printed baseplate above the lens. First, a second GRIN lens was optically glued (Norland Optical Adhesive 68, Edmund Optics) to the surface of the implanted lens and cured with UV light. The pitch of each GRIN lens is 0.25, so implanting 2 together provides a 0.5 pitch. This half pitch provides translation of the image at the bottom surface of the lenses to the top while maintaining the focal point below the lens. This relay implant enables access to tissue deep below the skull surface. Magnification of the image is performed by the miniscope. The miniscope was placed securely in the baseplate and then mounted to the stereotax to visualize the calcium fluorescence and tissue. The baseplate was then cemented in place above the relay lenses at the proper focal plane and allowed to cure. Once rats had been baseplated, they were placed on food restriction to reach a goal weight of 85% *ad lib* weight and then began behavioral training. Time between the beginning of the surgical procedures and the start of behavior training was typically 6–8 wk.

## Scopolamine Injections

On some training days rats were injected intraperitoneally with vehicle (0.9% sterile saline) or scopolamine (1 mg/kg dissolved at a concentration of 1 mg/mL) prior to shock sessions. After the injection, rats were returned to their home cage for 30 min (located on a cart in the experiment room) for 30 min prior to starting the session.

## Histology

At the end of the experiment, rats were anesthetized with isoflurane, intraperitoneally injected with 1 mL of pentobarbital, then transcardially perfused with 100 mL of 0.01 M PBS followed by 200 mL of 4% paraformaldehyde in 0.01 M PBS to fix the brain tissue. Brains were sectioned at 40 μm thickness on a cryostat (Leica), mounted on gelatin prepared slides, then imaged on a confocal microscope (Zeiss) to confirm GFP expression and GRIN lens placement (see examples in *Figure 2—figure supplement 1*).

## MiniLFOV calcium imaging system

To record calcium activity during unrestrained behavior, we utilized a large-field-of-view version of the Miniscope imaging system, 'MiniLFOV' (*Guo et al., 2023*). This open-source epifluorescence imaging camera weighs 13.9 g with a 3.6 × 2.3 mm field of view while maintaining a 2.5 um resolution at the center of view and 4.4 um at the periphery. The MiniLFOV can record at 22 Hz, has a 5 MP CMOS imaging sensor, an electrowetting lens for digitally setting the focal plane, and a modular lens configuration to enable a longer working distance (either 1.8 mm, used here, or 3.5 mm). The system is 20× more sensitive than previous v3 Miniscopes and twice as sensitive as current v4 Miniscopes. Power, communication, and image data are packed into a flexible 50 Ω coaxial cable (CW2040-3650SR, Cooner Wire) using power-over-coax filtering and a serializer/deserializer pair for bidirectional control communication with I²C protocol and unidirectional high bandwidth data streaming. The MiniLFOV

device interfaces with UCLA open-source Miniscope DAQ Software to stream, visualize, and record neural dynamics and head orientation data. This DAQ platform and software allow for excitation intensity adjustment, focus change, image sensor gain selection, and frame rate setting. See the MiniLFOV website (github.com/Aharoni-Lab/Miniscope-LFOV) and methods paper (*Guo et al., 2023*) for further details and printable part files.

## Trace extraction and spike inference

To accelerate image processing, each session's video stack was cropped to a rectangle that excluded edge regions containing no neural activity in any session from that animal. The stack was also temporally downsampled to every other frame, yielding an effective sample rate of ~11 Hz. Non-rigid motion correction was applied to remove residual non-uniform tissue motion artifacts (*Pnevmatikakis and Giovannucci, 2017*). Source extraction was then performed using the Python implementation of the Calcium Imaging Analysis package, 'CaImAN' (*Giovannucci et al., 2019*), yielding individual spatial contours and demixed temporal fluorescence traces for each detected neuron; parameters used for source extraction can be found in scripts provided at https://github.com/tadblair/tadblair/tree/Blair_et_al. Deconvolved spikes were derived from denoised fluorescence traces by CaImAN 's 'deconvolveCa' function using a second-order autoregressive model with automated estimation of baseline and convolution kernel parameters. To match cell contours across sessions, spatial contour weights from CaImAn were thresholded to 50% of their peak value to generate binary pixel masks that were then analyzed using *CellReg* (*Sheintuch et al., 2017*) for matching contours across all sessions included in the analysis for each rat.

## Behavior tracking

A webcam mounted in the behavior room tracked a red LED located on the top of the miniscope, and this video was saved alongside the calcium imaging via the miniscope software with synchronized frame timestamps. These behavior video files were initially processed by custom Python code, where all the session videos were concatenated together into one tiff stack, downsampled to 15 frames per second, the median of the stack was subtracted from each image, and finally they were all rescaled to the original 8-bit range to yield the same maximum and minimum values before subtraction. Background subtracted behavior videos were then processed in MATLAB. The rat's position in each frame was determined using the location of the red LED on the camera. Extracted positions were then rescaled using an inverted model of the empirically measured camera distortion and convert the pixel position to centimeters according to the maze size. Positional information was then interpolated to the timestamps of the calcium imaging video.

## Behavioral performance measures and criteria

During the initial phase of training where rats learn to prefer the short path over the long path on the maze, rats were required to reach a criterion of earning at least two rewards per minute by taking the short path and earning at least twice as many rewards on the short versus long path during a single 15 min session. After reaching this criterion, rats were given avoidance training in the next session as explained in 'Results.' After a single-avoidance training session, rats received standard (non-shock) sessions until avoidance behavior was extinguished. Rats were considered to have extinguished avoidance behavior on the first post-training day that they recovered to the original criterion (defined above) for preferring the short path.

48 hr retention of short path avoidance was quantified by the ratio $A = N_{+48}/N_{-48}$ , where $N_{+48}$ and $N_{-48}$ are the number of rewards earned on the short path during the first session after and before the training session, respectively. Hence, lower values of $A$ indicated better 48 hr retention of short path avoidance. In the drug-free shock condition, an individual rat was considered to show failed 48 hr retention of short path avoidance if $A > 0.6$, which was the cutoff for the top quartile of $A$ values in all rats (n = 11) from which imaging was successfully obtained during the drug-free shock condition (of the 14 rats that received drug-free shock training, one was omitted from imaging analysis because shocks were mistakenly delivered at the start of the session, and two were omitted because they failed to ever reach criterion for extinction of avoidance behavior after multiple post-training sessions). In the scopolamine shock condition, an individual rat was considered to show successful 48 hr retention of short path avoidance if $A < 0.31$, which was the cutoff for the bottom quartile of $A$ values in all rats

(n = 9) from which imaging was successfully obtained during the scopolamine shock condition (of the 10 rats that received scopolamine shocks, 1 was omitted from imaging analysis because the GRIN lens shifted and imaging was lost prior to the avoidance training session).

## Behavioral downsampling

Since behavior on the maze was altered by training and drug administration (*Figure 1*), imaging analyses only included data from beeline trials during which rats ran from one end of the track to the other with no change in direction (a beeline trial was defined as a continuous traversal of the short path throughout which the rat's 1D position along the track changed monotonically in one direction). This assured that rats were behaving as similarly as possible during analyzed data segments from all sessions. Beeline trials from each session were then further downsampled prior to imaging analyses to control for potential effects of behavior or sampling differences on the outcome of imaging analyses.

For comparisons of pre-training versus training sessions (*Figure 2C–H*) in the drug-free versus scopolamine shock conditions, it was necessary to correct for two imbalances in behavior between the drug-free versus scopolamine conditions (*Figure 2—figure supplement 2A*): (1) rats ran more slowly (and thus also ran fewer beeline trials) during training sessions on scopolamine than drug-free, and (2) rats ran more pre-training trials per session in the scopolamine than the drug-free condition. To correct for imbalance #1, we computed that the mean number of beeline trials (in the LR and RL directions combined) during training in the scopolamine condition (23.85 trials) was 64% of mean beeline trials in the drug-free condition (37.22 trials); since running speed was also slower for scopolamine training, we sorted each rat's drug-free training beeline trials in order of running speed and omitted the fastest 36% of trials from each running direction (rounded to the nearest integer). This simultaneously equalized beeline trial counts and running speeds during training sessions between the drug-free versus scopolamine conditions. To correct for imbalance #2, we computed that the mean number of beeline trials during pre-training (in the LR and RL directions combined) in the drug-free condition (53.78 trials) was 71% of mean beeline trials in the drug-free condition (76.0 trials); we omitted the last 29% of trials from each running direction in the pre-training session. This equalized trial counts across training conditions without significantly affecting mean running speeds. Also, since omitted pre-training trials were removed from the end of the session, the effective duration of pre-training sessions was reduced from 15 min down to a shorter duration that was more similar to the 10 min pre-shock period during which beeline trials were sampled from the training session. After these behavioral downsampling measures had been taken, rats in both conditions still ran fewer beeline trials during the analyzed portion of the training session (the first 10 min prior to shock delivery) than during the included portion of the pre-training session as a consequence of the fact that scopolamine rats ran fewer trials in the training session and this low trial count had to be matched in the drug-free group. But the difference in trial counts between pre-training and training sessions, though large, was similar for the drug-free and scopolamine training conditions, and was therefore assumed not to be a confound in analyses of the training condition effects (*Figure 2C–H*).

For comparisons of pre-training versus post-extinction sessions (*Figure 3*), it was necessary to correct for two imbalances in behavior between the drug-free versus scopolamine conditions (*Figure 2—figure supplement 2*): (1) rats in the drug-free shock condition tended to run fewer laps at lower speeds during the post than pre session (presumably because they still had some residual fear of the short path even after reaching the extinction criterion), whereas rats in the barrier condition tended to run more laps at higher speeds during the post than the pre session (presumably because they continued to acquire a preference for the short path without being interrupted by avoidance learning), and (2) during both the pre and post sessions, rats in the scopolamine shock condition ran more short path beeline trials than rats in the other two conditions. Imbalance #1 was corrected in two steps. The first step was to take the difference between the number of pre and post session trials for each rat in the drug-free condition: $N_{diff}^{DF} = N_{pre}^{DF} - N_{post}^{DF}$. Trials from the pre session were then sorted by speed, and the fastest $N_{diff}^{DF}$ trials were removed from the pre session, which equalized the number of pre and post session beeline trials while also making their mean running speeds much more similar, so that there was no longer a significant running speed difference (*Figure 3B*, right). The second step was to take the difference between the number of pre and post session trials for each rat in the barrier condition: $N_{diff}^{BAR} = N_{post}^{BAR} - N_{pre}^{BAR}$. Trials from the post session were then sorted by speed, and the fastest $N_{diff}^{BAR}$ trials were removed from the post session, which equalized the number of pre and

post session beeline trials while also making their mean running speeds much more similar, so that there was no longer a significant running speed difference (*Figure 3B*, right). Then an additional 20% of the fastest trials were removed from both sessions to render the mean number of beeline trials per session similar across the drug-free and barrier conditions. To correct for imbalance #2, the absolute difference was taken between the number of pre and post session trials for each rat in the drug-free condition: $N_{diff}^{SCOP} = \left| N_{pre}^{SCOP} - N_{post}^{SCOP} \right|$; trials from the scopolamine shock condition were then sorted by running speed, and the number of pre and post trials was equalized by removing fastest $N_{diff}^{SCOP}$ trials from whichever session had fewer trials in a given rat. Then an additional 25% of the fastest trials were removed from both sessions to render the mean number of beeline trials per session similar across the drug-free and scopolamine shock conditions.

## Spatial tuning properties

To analyze acute effects of scopolamine on spatial tuning properties, we measured each peak Af/s rate, out-of-field Af/s rate, tuning width, and spatial information content from place cell tuning curves. Peak Af/s rate was defined as the maximum Af/s rate observed for any of the 23 bins in the tuning curve. Out-of-field Af/s rate was defined as the mean rate in the subset of spatial bins that had rates <50% of the peak Af/s rate. Tuning width was computed as $W \times N_{50}$ , where $W$=10.8 cm is the width if each bin and $N_{50}$ is the number of bins that had rates >50% of the peak Af/s rate. Spatial mutual information content was measured in units of bits per active frame (bits/Af), which was computed as (*Skaggs et al., 1993*)

$$-r_B log_2 \left( \underline{r_B} \right) + \sum_{i=1}^{B} r_i p_i log_2 \left( r_i \right)$$

where $B$ is the set of spatial bins with non-zero mean Af/s rates, $r_i$ is the mean Af/s rate in spatial bin $i$, $p_i$ is the probability that the rat was occupying bin $i$, and $\underline{r_B}$ is the mean Af/s rate over all bins in the set $B$.

## Selection of place cells

Stable place cells were identified in each imaging session by randomly splitting the session's beeline trials (separately for each running direction) into two halves, and then computing two spatial tuning curves, one for each randomly chosen half of the beeline trials. A Pearson correlation coefficient ($R$) between these two tuning curves was then computed, and its p-value was recorded. A cell was classified as a place cell in a given session if the median p-value for $R$ was <0.01 over 200 iterations of random beeline splits in either of the two running directions. To be classified as a place cell, a neuron also had to generate an average of at least one inferred spike per beeline trial and was not permitted to fire at >70% of its peak inferred spike rate over more than 50% of the track (this was done to exclude likely interneurons and thereby restrict analysis to putative pyramidal neurons).

## Recurring place fields

A recurring place field was defined as an LR or RL tuning curve that (1) met the spatial selectivity criteria described above during at least one of the two sessions and (2) contained active frames (i.e., generated at least one inferred spike) during subsampled beeline trials in the specified running direction from both sessions. Place cell tuning curves were excluded from the between-session heatmap pair if they contained *non-recurring place fields*, defined as those that (1) met spatial selectivity criteria in during one of the two sessions, but (2) generated no inferred spikes (and thus went completely undetected) during subsampled beeline trials in the specified running direction from the other session.

## Shock responses

To analyze shock-evoked responses of CA1 place cells, we identified each frame during which the rat made an entry into the electrified shock zone and noted in which direction the rat was running when shock onset began. For each entry into the shock zone, a significant shock-evoked response occurred when the count of active frames in a 2 s window after entry into the electrified shock zone exceeded the confidence limit (p<0.01) for a Poisson fit to the baseline distribution of active frame counts from prior visits (during the first 10 min of the session) to unelectrified shock zone in the same running

direction. Most rats experienced the shock in both running directions before they began to avoid the short path, but a few rats only experienced the shock in one direction, and in a small number of rats, more than two shocks were received before the rat began avoiding the short path. To equalize statistical power for detecting shock responses across all rats (regardless of the shocks they received), a neuron was classified as shock-responsive only if it exhibited a significant shock response during the rat's first or second visit (or both) to the electrified shock zone; later visits to the shock zone were not considered, so that place cells would not have a higher likelihood of being classified as shock-responsive in rats that made more visits to the shock zone.

To compute population averaged responses of place cells to shocks, only the first significant shock-evoked response of each shock-responsive place cell (from either the first or second shock encounter, whichever yielded the first significant response) was included in the average, so that each cell's contribution to the average always came from a single significant shock response trial, and therefore did not vary depending upon the number of shocks different animals received. For shock-responsive place cells, the population average included only the first shock encounter (which by definition failed to elicit a significant shock response), so that again, each cell's contribution to the average came from a single trial.

Distributions of tuning curve peak locations (*Figure 4A*, right) for shock-responsive place cells included only tuning curve peaks for the running direction in which the rat received its first significant shock-evoked response (from either the first or second shock encounter, whichever yielded the first significant response). Distributions of tuning curve peak locations for shock-responsive place cells included tuning curve peaks from both running directions if the rat received its first and second shock in different directions, but if the first and second shock were both received in the same running direction, then only the tuning curve peak for that direction was included in the distribution of peak locations.

To assess whether a place cell's tuning curve peak was on the approaching versus departing side of the shock zone, the 23-bins of the tuning curve were reflected across the middle (bin 13) for RL tuning curves, but not for LR tuning curves, so that for both running directions, tuning curve peaks in bins 1–12 were on the approaching side of the shock zone, and peaks in bins 14–23 were on the departing side of the shock zone. For tuning curves with peaks in the center (bin 13), an additional step was taken to measure the percentage of the area under the tuning curve on either side of center. If the area in bins 1–12 was greater than in bins 14–23, then the tuning curve peak was considered to be on the approaching side; otherwise, it was considered to be on the departing side.

## Position decoding

For decoding analysis, frame positions in one imaging session were predicted using place cell heatmaps derived from a different session. For each analyzed pair of pre- and post-sessions, decoding was performed using memoryless Bayesian inference (*Davidson et al., 2009*; *Zhang et al., 1998*) with place maps constructed from each running direction. Post-training positions were predicted from pre-training heatmaps (pre→post), and vice versa (post→pre), taking the maximally likely position bin at each frame, and results from both directions were averaged together to obtain the results in *Figure 3G*. A subsampling procedure was used to equalize the number of place cells used for each running direction (LR and RL, 100 subsample iterations), so that any observed differences in decoding performance were not attributable to differences in the sizes of the population vectors (i.e., the number of recurring place cells) that were used for pre-training versus cross-training decoding.

The *decoding error* (DE) in each frame was computed as the absolute error (in cm) between the rat's actual versus predicted position in that frame, DE = abs($x_{act}$-$x_{pred}$). The *shuffled decoding error* ($DE_{shuffle}$) was derived in the same manner, except that place cells in the population vector were randomly permuted against their place maps before deriving the predicted position, $x_{pred}$. Shuffled decoding error was computed multiple times per frame using different random place cell permutations (n = 2000 shuffles). The decoding error in a given position bin was measured as the median of the absolute difference (in cm) between the rat's actual versus predicted position over all visits to that bin. The non-shuffled decoding error was subtracted from the median $DE_{shuffle}$ to obtain the *accuracy gain* in each frame, $G$ = median($DE_{shuffle}$)-DE, which measured by how many centimeter the decoding error was reduced when position was predicted from non-shuffled rather than shuffled population vectors. The median value of $G$ was derived for all frames in each actual position bin,

$x$, yielding an accuracy gain measurement at each position on the track, $G(x) = \text{median}(G_{x1}, G_{x2}, \ldots G_{xn})$. Accuracy gain was computed separately at each position in each running direction (LR and RL) and each decoding direction (pre→post and post→pre) to obtain four different $G(x)$ values per session pair: $G_{pre \to post}^{LR}(x)$, $G_{pre \to post}^{RL}(x)$, $G_{post \to pre}^{LR}(x)$, and $G_{post \to pre}^{RL}(x)$. For statistical analyses, each of these four values was treated as an independent observation at each position. Accuracy gains tended to be greater near the ends of the track than in the middle, but this was mainly a consequence of bias in the $G(x)$ accuracy measure, rather than better decoding at the ends of the track, because greater reductions of $G(x)$ were attainable near the ends of the track, where $x_{act}$-$x_{pred}$ could span the entire track length, than in the middle, where $x_{act}$-$x_{pred}$ could only span half the track length at most. To eliminate this positional bias, we used Wilcoxon signed-rank tests (rather than subtraction) to test whether the median error difference, $DE_{shuffle}$-$DE$, differed significantly from zero (see 'Methods'). p-Values from the Wilcoxon tests provided a measure of decoding performance that was unbiased by track position, but these p-values were non-normally distributed, and thus could not be analyzed to test for interaction effects using parametric statistics. We therefore took the negative log of the Wilcoxon p-value and multiplied by the sign of $G$ to derive a *decoding accuracy index*, $DAI = -\log_{10}(P) \times \text{sign}(G) - 2$, which was both normally distributed and unbiased by track position. Subtracting 2 at the end of the expression was done so that $DAI = 0$ would correspond to $p=0.01$ for more significant decoding from unshuffled than shuffled population vectors since $\log_{10} 0.01 = 2$.

## Acknowledgements

We thank Dr. Andrew Howe, Shiyun Wang, Ryan Grgurich, Umais Khan, and Lorelei Mae Cordes for helpful comments on the experimental design, data collection, and manuscript. This work was supported by NSF NeuroNex 1704708 (HTB, PG, DA) and RO1-MH062122 (MSF).

## Additional information

### Funding

| Funder | Grant reference number | Author |
|---|---|---|
| National Science Foundation | NeuroNex 1704708 | Hugh T Blair Peyman Golshani Daniel Aharoni |
| National Institute of Mental Health | RO1-MH062122 | Michael S Fanselow |

The funders had no role in study design, data collection and interpretation, or the decision to submit the work for publication.

### Author contributions

Garrett J Blair, Conceptualization, Data curation, Software, Formal analysis, Validation, Investigation, Visualization, Methodology, Writing – original draft, Writing – review and editing; Changliang Guo, Conceptualization, Software, Investigation, Methodology, Writing – review and editing; Shiyun Wang, Collection of supplementary behavioral data; Michael S Fanselow, Conceptualization, Resources, Funding acquisition, Writing – review and editing; Peyman Golshani, Supervision, Funding acquisition, Project administration, Writing – review and editing; Daniel Aharoni, Conceptualization, Resources, Software, Supervision, Funding acquisition, Validation, Methodology, Writing – review and editing; Hugh T Blair, Conceptualization, Resources, Data curation, Formal analysis, Supervision, Funding acquisition, Validation, Writing – original draft, Project administration

### Author ORCIDs

Garrett J Blair https://orcid.org/0000-0003-2724-8914
Michael S Fanselow http://orcid.org/0000-0002-3850-5966
Daniel Aharoni http://orcid.org/0000-0003-4931-8514
Hugh T Blair http://orcid.org/0000-0001-8256-5109

## Ethics

All experimental procedures were approved by the Chancellor's Animal Research Committee of the University of California, Los Angeles, in accordance with the US National Institutes of Health (NIH) guidelines. protocol #2017-038.

## Decision letter and Author response

Decision letter https://doi.org/10.7554/eLife.80661.sa1
Author response https://doi.org/10.7554/eLife.80661.sa2

# Additional files

## Supplementary files
• MDAR checklist

## Data availability

Source data and code for reproducing the figures are available at: https://github.com/tadblair/tadblair or https://doi.org/10.5068/D1ZT2S.

The following dataset was generated:

| Author(s) | Year | Dataset title | Dataset URL | Database and Identifier |
|---|---|---|---|---|
| Blair GJ, Guo C, Wang S, Fanselow MS, Golshani P, Aharoni D, Blair HT | 2023 | Hippocampal place cell remapping occurs with memory storage of aversive experiences | https://doi.org/10.5068/D1ZT2S | Dryad Digital Repository, 10.5068/D1ZT2S |

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
