## [Editor Report]

This paper describes important results obtained from multi-cellular imaging of CA1 cells using large-field-of-view miniscopes in rats performing a shock avoidance task. By exploiting behavioral (barriers) and pharmacological (scopolamine) manipulations the authors provide convincing results on cell remapping dynamics during aversive learning. This work will be of interest for the neuroscience community by setting new methodological standards and providing data for across species comparisons.

---

## [Decision Letter]

**Decision letter after peer review:**

Thank you for submitting your article "Disruption of place cell remapping by scopolamine during aversive learning" for consideration by *eLife*. Your article has been reviewed by 3 peer reviewers, including Liset M de la Prida as Reviewing Editor and Reviewer #1, and the evaluation has been overseen by Laura Colgin as the Senior Editor. The following individual involved in the review of your submission has agreed to reveal their identity: Alexandra T Keinath (Reviewer #3).

Essential revisions:

This paper describes results obtained from multi-cellular imaging of CA1 cells using large-field-of-view miniscopes in rats performing a shock avoidance task. By exploiting behavioral (barriers) and pharmacological (scopolamine) manipulations, the authors explore cell remapping dynamics during aversive learning. We very much value using large-field-of-view miniscope imaging to provide place cell data from rats, thus favoring inter-species comparisons. We also appreciate the inclusion of an appropriate set of control analyses and experiments to exclude potential confounds, especially when it comes to comparison between groups. While all reviewers agreed that the work will be of interest, there were also some concerns requiring substantial revision.

A point of discussion was whether the results of scopolamine are novel and specific enough to support claiming a direct effect on memory processes. There were also concerns regarding additional methodological controls which are required. We feel they can be addressed with additional analyses and experiments. In a consultation session, we agreed the following revisions are required before a definitive recommendation can be made:

1) Control experiments for scopolamine are required. These include:

a) A control is needed to show that the injection procedures do not impair general behaviour (e.g. motivation, attention, level of stress) as well as other forms of learning;

b) Consider effects in the absence of shock learning. To be able to claim that the effects of scopolamine are specific to aversive learning, a control with either no learning or appetitive learning (e.g. of a reward location, which also involves place field reorganization in some cases) would be useful. We feel that acknowledging the systemic/brain-wide/unspecific effect of scopolamine is not enough and you should provide some additional evidence of specificity.

c) Consider behavioral side effects of a systemic approach. Similarly, consider that functional effects of scopolamine (power and coordination of theta oscillations brain-wide) can additionally provide an explanatory value of the effects. Some of these are discussed but some others need support.

2) Additional control analyses are required, including:

a) Population vector analysis should be tested against shuffled distributions.

b) Population-level analysis in terms of the percentage of cells meeting place-cell criteria across manipulations; the equivalency of place cell measures when including all principal cells, not just those meeting (same-session) pre-selection criteria.

c) There are also some issues with the different pre-/post-training timing (8 days vs 6 days after pre-training) used in remapping effects with/without scopolamine that require clarification.

3) Clarification regarding LFOV:

a) Using large GRIN lenses may bring undesired histological damage and we wonder whether flattening the dorsal hippocampus may have consequences on functional readouts.

b) Given improvements in LFOV imaging with your approach, we wonder whether center/periphery ROI imaging may be or may not be affected. Given individual variability of imaging ROIs per rat, we would like to see this more carefully addressed.

4) Regarding impact and novelty, you may consider the following suggestions:

a) We discussed whether the above revisions may offer a chance to reinforce the novelty of your observations, by presenting pharmacological and behavioral (barriers) experiments as manipulations to better focus on the study of place cell remapping dynamics during aversive learning in rats. Actually, the abstract poorly integrates barrier results, which we feel should be mentioned. The authors may consider whether by highlighting/refocusing around some of these angles their manuscript would gain additional impact and broader interest.

b) Consider also tuning the title accordingly if required. Please, note that the title and/or the abstract should provide an indication of the biological system under study.

c) We also discussed whether using LFOV imaging in rats is an important addition to your paper and we would ask you to consider expanding the value of the approach when it comes to imaging from larger brains such as in rats versus mice. We feel in addressing questions regarding GRIN lenses and image aberration you may find the opportunity to highlight this better.

Please see the comments from each reviewer for more details.

*Reviewer #1 (Recommendations for the authors):*

Here is a list of comments for the authors to improve their work:

1. The body of the analysis is centered on the evaluation of population vector correlation matrices between behavioral phases and conditions. To build them, the authors rely on tuning curve heatmaps calculated from place cells in a reference condition (pre). One major problem with population tuning curves is that their typical 'diagonal' shape is very sensitive to noise so all curves and derived analysis should be tested against shuffled distribution as well. This affects data in Figure 2H,I; Figure 3D-F; Figure 4B,C.

2. The authors nicely control for their main figure analysis using the subset of scopolamine-treated rats that retained avoidance, but they do not provide data on the similarly important group of drug-free rats that DO NOT retain avoidance. They may help to provide additional controls for all main figure analyses as well. In addition, the paper would benefit from examining whether remapping indices can explain the behavioral performance of all rats, without resorting to categorizing them by performance level (i.e. using correlation analysis and/or predictive model analysis).

3. Using large GRIN lenses for large-field-of-view is a major strength of the paper when applied to imaging from the rat hippocampus. However, this can yield undesired histological damage. For instance, pushing the lens against the alveus for image flattening results in a reduction of the stratum oriens and atrophy of basal dendrites. This is inevitable and should not be considered by the authors as a criticism, but worth mentioning/controlling for effects. Are any of the behavioral and/or functional readouts correlated with some histological damage?

4. Similarly, there is natural variability in imaging from cells at the center/periphery of the GRIN, which differ in aberration effects. Since there is a different contribution across animals another useful control is to evaluate the consistency of conclusions for data obtained from center/periphery within and between animals. This would also help to set standards for the field, especially for LFOV imaging. Importantly, there is poor information on how the ROIs are defined, what methods are used for segmentation and how imaging from different sessions are treated (same ROIs, different ROIs, etc…), and what signals are evaluated dF/F with/without denoising, models of spike inference, etc…

*Reviewer #2 (Recommendations for the authors):*

Here are some detailed cpmmrnts.

1) Please add page and line numbers to ease review.

2) Given that several of the rats did the drug-free shock learning twice, how were these data analysed? Was only one of the sessions per rat chosen to contribute to the analysis, if yes, how?

3) Was there an influence of the order of sessions on the results (e.g. scopolamine during the first shock session, the second one, after a barrier session or not)?

4) Any analysis of the number of trials, if comparing across sessions of different durations, should be done on a number of trials/minute to allow for cross-sessions comparisons (e.g. Sup Figure 2 A).

5) Analysis of remapping: remapping is not the same as instability. It corresponds to a consistent activation of place fields in specific locations, that change (in a consistent manner) in a different condition, with the activity pattern, mostly reinstated when coming back to the original condition. Otherwise, changes in place cell firing that are not consistent or reproducible are considered drift or instability of the spatial firing. It is unclear if the current paradigm and analysis tests remapping and not simply instability.

6) In the 'Scopolamine does not disrupt place cell responses to shock' section: individual place cells are considered to 'respond' to the shock if their activity increases during entry into the shock zone. However, place cells can also respond by decreasing their activity, or even changing their place field shape or location, but this is not analysed.

7) Given that place cells can modulate their firing spontaneously with time (mentioned by the authors), any analysis of place cells between two different conditions where time has elapsed should have as a control a similar comparison, without the parameter of interest being changed, but with approximately the same amount of time passed or at least number of sessions (as the authors do in one of their analyses).

8) Nomenclature: In general, it would be good to avoid the use of abbreviations as much as possible, including in the figures, to help readability. Also, the different terms used (mostly condition or session names) can be confusing and there may be errors in figures or legends that make the interpretation difficult. The use of pretraining, training, and post in some instances, but pre10 post 5 in others (within the same session), is also a bit confusing. Perhaps the 10 min part of the shock training sessions that is pre-shock should be called the normal session and then the shock session would be called the shock session (or barrier session), without lumping both in the same session. Also, there are 2 forms of training: training to choose the shortest path, then training of the shock – and then extinction of the shock learning. Perhaps the authors could find a more intuitive and consistent way to name their sessions. What is 'post-extinction' (e.g. in 'post-extinction CA1 population vectors)? Is it referring to the activity during the extinction sessions? Or during the sessions that followed the extinction sessions, which according to Figure 2B will always be a shock session? It seems that this is sometimes called post-training (e.g. Figure 1J) and sometimes post-extinction (e.g. Figure 1J, legend). Is the place field recurrence rate (Figure 2G) the same as place field recurrence ratio (in results)? Do subsampled and down sampled mean the same here?

9) Related to this, in several instances the axis labels or legends of a figure are not self-sufficient and could be enhanced with more information about the measure that is shown (e.g.: Figure 1J: how is %baseline computed?).

10) The graph in Figure 1B seems to show that after each shock session, the rats have 1 extinction session. But in the main text, it is said that 'rats continued to run standard sessions on the maze every 48h until they reached the avoidance extinction criterion", which seems to assume that this would take several sessions. It seems that the graph should be adapted to reflect how many sessions were run. It is not clear what happens after the barrier sessions – do the rats move on to the shock learning sessions immediately? Does the '3s' mean that they do instead of the barrier session 3 times, or maybe extinction sessions?

11) It is a bit unclear what the -48/+48 measure means on some of the figure axes and in text. Does that mean that what is compared is data from 4 days apart? Or just 2 days apart? If it's the latter, shouldn't the measure be 0-48h or -24/+24?

12) Rejecting specific trials from the analysis does not change the behaviour of the rats during data collection; it only makes the analysis more balanced. The sentence "This assured that rats were behaving as similarly as possible during data collection in all sessions" should be rewritten to reflect this.

13) Figure 2E could show proportions out of the total number of putative pyramidal cells analysed, not just out of detected place cells, to see if the percentage of active place cells changes.

14) Tuning curve 'sequences' (showing the tuning curves of all cells ordered according to peak firing) are used in many of the plots, but it is unclear what information to take from them. Reorganizing the figures to emphasize the quantification of the results might be useful and perhaps removing some of the sequence plots unless they convey a specific message or can be used as an example.

15) Related to this, most of the figures are really small. Focusing on the ones that provide quantitative results (i.e. less 'tuning curves' plots) and making those bigger would probably help the reader.

16) Descriptions of some specific procedures could be added in the methods (listed in no specific order):

a) How were the beeline trials selected?

b) How were putative spikes extracted/reconstructed from the calcium imaging signal (deconvolution?)?

c) How was the injection procedure performed, and where were the rats kept after it? Were there any observable effects on the behaviour?

d) what was the weight/age of the rats at the start of the experiment?

e) what is the statistical significance threshold chosen?

f) how were the shocks delivered and calibrated? Did they not generate artefacts in the imaging?

17) The difference between past findings and the novelty of the present study could be explained better (for example, Sun et al. 2021 seems very relevant to the current study but is practically not discussed, also, Wu et al. (2017) [Hippocampal awake replay in fear memory retrieval. Nature Neuroscience 20, 571-580] and Girardeau et al. (2017) [Hippocampal awake replay in fear memory retrieval. Nature Neuroscience] seem like they should be discussed).

18) How were the A>0.6 threshold and A<0.31 threshold chosen?

19) The authors have selected for analyses of the 'control' (non-scopolamine) group only rats that learn to avoid the shock zone, and for analyses of the 'test' (scopolamine) group the rats that do not learn. Possibly related to this, it seems that the different groups have different behaviours in the pre-scopolamine session. I believe this is not the right approach; instead, if anything, the groups that are going to be compared should be equalized in as many parameters as possible at least prior to the manipulation of interest. If anything, the variability within each group should be used to assess the correlation between remapping/ instability of place cells and performance.

*Reviewer #3 (Recommendations for the authors):*

On the whole, I very much enjoyed reading this work, I found the results to be quite interesting and the presentation to be clear. I have only one question that can be addressed with further analysis. An essential claim of the manuscript is that the spatial codes measured during the drug-free and scopolamine shock sessions are equivalent in terms of within-session quality (and I appreciated all of the subsampling controls to match gross behavioral measures), yet I think this is only shown to be the case for 'place cells' which were preselected on the basis of various within-session measures. Can you confirm that: (1) the percentage of cells meeting place-cell criteria is equivalent across the two manipulations (not just the raw number of cells), and that (2) the equivalency of these measures holds when including all principal cells, not just those meeting (same-session) preselection criteria? I would be convinced about this claim if this were the case.

[Editors’ note: further revisions were suggested prior to acceptance, as described below.]

The manuscript has been improved but there are some remaining issues that need to be addressed. Please, check the reviewers' comments and correct/make changes to address the remaining comments. Regarding R2 comments (previous comment 6) asking for additional analysis, please feel free to make a decision on whether this can be addressed textually.

*Reviewer #2 (Recommendations for the authors):*

Hippocampal place cell remapping occurs with memory storage of aversive experiences.

Tracking no: 29-05-2022-RA-*eLife*-80661R1

The scientific approach and interpretation of this revised manuscript have been substantially updated (mostly in the introduction and discussion) and appear much more straightforward and supported by the results. The authors have answered most of my comments and added a behavioural-only control for the injection procedure, which is very useful (although the results for the control group might need some clarification, see comment below). Overall, I find the manuscript to be much improved and the results very interesting.

Comments related to previous comments:

Previous Comment 4: Normalizing trial counts: Currently, the data shown in Figure 2 – supplement 2 in the 'trn' group is taken from 10 minutes of behaviour, while data in the other groups are obtained from 15-minute sessions. Thus, the figure gives the impression of the trial number decreasing between the pre and trn sessions, but this is due (at least partially) to the shorter session duration for trn. This is why I was suggesting using trials/min in order to see if the rats still run fewer trials for an identical duration. If the authors prefer to keep the absolute number of trials as a measure, I suggest adding in the figure itself a clear mention of the duration of data included in each group to avoid misunderstandings.

Previous Comment 5 Analysis of remapping vs instability: I believe the authors have actually done an analysis that answers this, by computing the within-session correlation of rate maps (showing that the cells are still spatially stable) in Figure 2I, by showing that the spatial information is similar between the pre and trn sessions even in the scopolamine condition, together with computing the pre vs post correlations; thus no action is needed.

Previous Comment 6 Response to shock by increased firing vs decreased firing: My suggestion would be to detect all active place fields contacting the shock zone in the pre10 session part, and analyze their firing in the post5 or subsequent session part. Using an active place field should prevent the low baseline problem described by the authors.

Previous Comment 16.e Choice of significance threshold: Sorry if the previous comment was unclear – I meant that the methods should specify which is the threshold for statistical significance chosen to be used throughout the manuscript to decide when a test result that produces a p-value is deemed significant vs not, e.g. 'tests were considered significant at p<0.05'.

New Comment 1 (NC1): Thank you for adding a behavioural control with injection of saline. I have some comments related to this new figure (Figure 1 – Supplementary figure 1):

a) Is data in the 'Shock' session from the entire 15 min or only the first 10 min pre-shock? I recommend showing separately pre10 and post5 to get a better understanding of the effects.

b) If all the data were included: why is it that the saline rats do not seem to avoid the shock zone during the shock session (compare pre and shock data in B or C)? This should be discussed in the main text.

c) Separating the data in the graph would also allow us to replicate the effect seen in the implanted rats where scopolamine rats avoid the short path even before the shock session. Does this effect underlie an increase in impulsivity, or impairment of decision-making, possibly linked to a decrease in general attention? Are such side-effects of the scopolamine known? Perhaps these types of possible side effects of the scopolamine could be added to the relevant discussion (e.g. in around L 596).

d) Ideally, some measure of general activity (average speed or total distance covered for the whole session, not just included trials) should be shown in the same way as the existing figures to further assess the effects of the scopolamine, injection procedures and shock on activity.

e) Why is the 'shock' session not called 'trn' as in Figure 2 Supplement 2?

NC2: L 24, L 52 (and also L 503) "We conclude that place cell remapping occurs in response to motivationally significant (rather than neutral) stimuli that are remembered rather than merely perceived and forgotten"; "studies have shown that place cells remap in response to behaviorally significant events": this is partially correct, however, these sentences and the corresponding one in the discussion imply that *only* behaviourally significant events can cause place cell remapping. But place cells also remap in response to events of low/no behavioural significance, e.g. a change in the odour/colour of the environment (Anderson and Jeffery 2003, J Neurosci; Jeffery et al., 2003) or events of intermediary behavioural relevance like the introduction of barriers in the absence of a navigation task (Rivard et al., 2004, Journal of General Physiology, Muller and Kubie, 1987, J Neurosci). These studies generally observe partial remapping (of a sub-population of the place cells) while perhaps the remapping in the shock condition in the current study is more global (but to test this, additional analysis of place fields away from the shock zone would be necessary). For these reasons, I suggest reformulating/removing the references to "motivationally significant (rather than neutral) stimuli" to perhaps focus on the remembered vs non-remembered aspect. Perhaps the *salience* of an event influences the degree of remapping throughout the place cell population… but that hasn't really been demonstrated by the current study.

NC3: L508 +: "Hippocampal place fields tend to be preferentially concentrated at locations where rodents spend large amounts of time […]" : this is not always the case (see for example Hok et al., 2007, J Neurosci; Pfieffer, 2022, J Neurosci), and might depend on specific task demands and behavioural stereotypy (reviewed in Nyberg et al., 2022). I suggest adding a clarification that the goal overrepresentation phenomenon is not ubiquitous and might depend on the task and navigational strategy.

NC4: L 536 'indicating that place fields shifted towards and away from the shocked location in equal proportions': could the authors point to a result showing this, instead of fields, for example, changing their size or firing rate?

---

## [Author Response]

Essential revisions:A point of discussion was whether the results of scopolamine are novel and specific enough to support claiming a direct effect on memory processes. There were also concerns regarding additional methodological controls which are required. We feel they can be addressed with additional analyses and experiments. In a consultation session, we agreed the following revisions are required before a definitive recommendation can be made:1) Control experiments for scopolamine are required. These include:a) A control is needed to show that the injection procedures do not impair general behaviour (e.g. motivation, attention, level of stress) as well as other forms of learning;

We have run an additional saline-only control group (along with additional scopolamine groups) to demonstrate that scopolamine does impair avoidance learning compared against saline-injected controls. As shown in the Supplement to Figure 1, rats receiving saline alone show significantly greater post-training avoidance of the short path than rats receiving scopolamine.

b) Consider effects in the absence of shock learning. To be able to claim that the effects of scopolamine are specific to aversive learning, a control with either no learning or appetitive learning (e.g. of a reward location, which also involves place field reorganization in some cases) would be useful.

We believe that the barrier control condition, in which rats were exposed to a neutral barrier rather than an aversive shock, is sufficient to address this concern. However, we agree with this and other reviewers that insufficient attention was given to this control in the original manuscript, so it is now mentioned in the abstract and considered more prominently in the introduction and discussion.

We feel that acknowledging the systemic/brain-wide/unspecific effect of scopolamine is not enough and you should provide some additional evidence of specificity.

We have added a paragraph to the discussion (ll. 665-678) in which it is explained why the main conclusions of our study (namely, that place cell cell remapping is related to storage of memories for aversive events) do not depend upon whether or not scopolamine’s pharmacological actions were localized to the hippocampus (as almost certainly they were not).

c) Consider behavioral side effects of a systemic approach. Similarly, consider that functional effects of scopolamine (power and coordination of theta oscillations brain-wide) can additionally provide an explanatory value of the effects. Some of these are discussed but some others need support.

These considerations have been thoroughly integrated into the discussion. We acknowledge that scopolamine likely affected hippocampal and non-hippocampal circuits (ll. 665-678) and discuss the drug’s acute effects upon behavior and place cell activity (ll.561-612), including citation of prior work on how mAChRs modulate theta oscillations (ll.580-589).

2) Additional control analyses are required, including:a) Population vector analysis should be tested against shuffled distributions.

New analyses show that all of the effects reported in the paper are abolished when the analyses are performed on shuffled data (Supplement 7 to Figure 2; Supplement 4 to Figure 3). Running this control analysis helped us to identify a minor flaw in the population vector correlation metric from the original manuscript (see reply to referee #1 comments below); this flaw has been corrected throughout the main text and figures by using a new population vector correlation metric referred to as place tuning stability (S). All previously reported effects survive the correction of converting to this slightly modified correlation measure.

b) Population-level analysis in terms of the percentage of cells meeting place-cell criteria across manipulations; the equivalency of place cell measures when including all principal cells, not just those meeting (same-session) pre-selection criteria.

Percentage of cells meeting place cell criteria are now reported in Figure 2D (right graph) and 3C (middle graph).

c) There are also some issues with the different pre-/post-training timing (8 days vs 6 days after pre-training) used in remapping effects with/without scopolamine that require clarification.

This appears to be a reference to an issue raised in a comment from referee #2. As explained in our reply to that comment below, we have controlled for this; the amount of time that elapses between sessions is identical for all analyses in Figure 2 (48 h) and very nearly so for all analyses in Figures 3 and 4, where all but 3 rats had identical intervals of 8d.

3) Clarification regarding LFOV:a) Using large GRIN lenses may bring undesired histological damage and we wonder whether flattening the dorsal hippocampus may have consequences on functional readouts.

We agree that the issue of tissue compression is a concern for GRIN lens experiments, but no more so for our current study than for numerous other studies that have used similar methods in mice and rats. We do not convinced that histological damage can be quantified in a reliable way from the micrographs, but we have acknowledged this issue by adding a sentence to the Results section (ll. 151-154):

“Although the GRIN lens caused some flattening and compression of stratum oriens (Supplementary Figure 1), we do not believe this caused significant disruptions of CA1 activity since spatial tuning properties of place cells were quite similar to those reported in prior studies of rats without lens implants.”

b) Given improvements in LFOV imaging with your approach, we wonder whether center/periphery ROI imaging may be or may not be affected. Given individual variability of imaging ROIs per rat, we would like to see this more carefully addressed.

Supplement 3 to Figure 2 now shows how spatial tuning and between-session recurrence of place cells depends upon the size (area in pixels) and eccentricity (distance from lens center) of cell contours in the imaging plane. Although there are small statistically significant effects of contour size and eccentricity upon the percentage of cells classified as place cells (and also on the probability that a place cell will recur from one session to the next), we not believe that these small effects or are of much interest or that they are likely to have impacted the outcome of our other analysis, for reasons discussed in the caption.

4) Regarding impact and novelty, you may consider the following suggestions:a) We discussed whether the above revisions may offer a chance to reinforce the novelty of your observations, by presenting pharmacological and behavioral (barriers) experiments as manipulations to better focus on the study of place cell remapping dynamics during aversive learning in rats. Actually, the abstract poorly integrates barrier results, which we feel should be mentioned. The authors may consider whether by highlighting/refocusing around some of these angles their manuscript would gain additional impact and broader interest.b) Consider also tuning the title accordingly if required. Please, note that the title and/or the abstract should provide an indication of the biological system under study.c) We also discussed whether using LFOV imaging in rats is an important addition to your paper and we would ask you to consider expanding the value of the approach when it comes to imaging from larger brains such as in rats versus mice. We feel in addressing questions regarding GRIN lenses and image aberration you may find the opportunity to highlight this better.

We are grateful for these suggestions and have made structural revisions to address them. The abstract (as well as the introduction and discussion) now address the barrier result and its importance for interpreting the findings. The title, introduction, and discussion have been thoroughly revised to shift emphasis away from implying that the scopolamine effects were end-goal of the study, and toward depiction of the scopolamine manipulation as a means to the more fundamental end of elucidating the role of remapping in storing memories of aversive experiences.

Please see the comments from each reviewer for more details.Reviewer #1 (Recommendations for the authors):Here is a list of comments for the authors to improve their work:1. The body of the analysis is centered on the evaluation of population vector correlation matrices between behavioral phases and conditions. To build them, the authors rely on tuning curve heatmaps calculated from place cells in a reference condition (pre). One major problem with population tuning curves is that their typical 'diagonal' shape is very sensitive to noise so all curves and derived analysis should be tested against shuffled distribution as well. This affects data in Figure 2H,I; Figure 3D-F; Figure 4B,C.

The requested controls against shuffled distributions are now reported in Supplement 7 to Figure 2 and Supplement 4 to Figure 3, which show results after circularly shifting inferred spikes against position tracking data during each beeline trial. When deconvolved spikes were shuffled prior to the analysis, spatial tuning and remapping effects were abolished, as expected. It is worth noting that in the course of performing this control analysis, we became aware of a minor flaw in the population vector correlation measure that we used in the original manuscript; this flaw has been corrected in the revised paper without major impact upon the reported results. Briefly, the flaw was as follows. We previously measured population vector correlations by taking the median R value in bins near the diagonal of the population vector correlation matrix. However, when running our analyses on shuffled distributions, we realized that this measure can become large not only when place cells exhibit similar spatial tuning across sessions, but also when non-place cells fire at similar baseline rates across the entire track during both sessions. The proper control for this is to subtract away the mean off-diagonal r-values (which tend to be small for place cells and large for non-place cells) in each individual correlation matrix before taking the median. Making this correction to our metric produced almost no differences from the outcomes of our population vector correlation analyses, because the analyses were already being performed upon tuning curves of place cells that had been selected for their spatial firing properties, and thus for the fact that they already had near-zero mean r-values off the diagonal (hence, subtracting these near-zero means under the new measure had minimal effect upon analysis outcomes). It does become important to perform this correction in the requested analyses of population vector correlation matrices computed from shuffled spike trains, since these matrices can have large values on the diagonal in the absence of spatial tuning. We thank the referee for proposing the shuffle control and thereby helping us to improve our population vector correlation measure.

2. The authors nicely control for their main figure analysis using the subset of scopolamine-treated rats that retained avoidance, but they do not provide data on the similarly important group of drug-free rats that DO NOT retain avoidance. They may help to provide additional controls for all main figure analyses as well. In addition, the paper would benefit from examining whether remapping indices can explain the behavioral performance of all rats, without resorting to categorizing them by performance level (i.e. using correlation analysis and/or predictive model analysis).

In the original manuscript, the results for drug free rats that did not retain avoidance were included in Supplementary Figures 4-7 (in the revised manuscript, these results are shown in Supplement 8 to Figure 2 and Supplement 3 to Figure 3). However, the reviewer is correct to point out that these results were not previously mentioned in the main text. We have addressed this by adding sentences to the main text that describe results for drug free rats that did not retain avoidance along with results for the scopolamine rats that did retain avoidance.

3. Using large GRIN lenses for large-field-of-view is a major strength of the paper when applied to imaging from the rat hippocampus. However, this can yield undesired histological damage. For instance, pushing the lens against the alveus for image flattening results in a reduction of the stratum oriens and atrophy of basal dendrites. This is inevitable and should not be considered by the authors as a criticism, but worth mentioning/controlling for effects. Are any of the behavioral and/or functional readouts correlated with some histological damage?

We agree that the issue of tissue compression is a concern for GRIN lens experiments, but no more so for our current study than for numerous other studies that have used similar methods in mice and rats. We do not convinced that histological damage can be quantified in a reliable way from the micrographs, but we have acknowledged this issue by adding a sentence to the Results section (ll. 151-154):

“Although the GRIN lens caused some flattening and compression of stratum oriens (Supplementary Figure 1), we do not believe this caused significant disruptions of CA1 activity since spatial tuning properties of place cells were quite similar to those reported in prior studies of rats without lens implants.”

4. Similarly, there is natural variability in imaging from cells at the center/periphery of the GRIN, which differ in aberration effects. Since there is a different contribution across animals another useful control is to evaluate the consistency of conclusions for data obtained from center/periphery within and between animals. This would also help to set standards for the field, especially for LFOV imaging.

Supplement 3 to Figure 2 now shows how spatial tuning and between-session recurrence of place cells depends upon the size (area in pixels) and eccentricity (distance from lens center) of cell contours in the imaging plane. Although there are small statistically significant effects of contour size and eccentricity upon the percentage of cells classified as place cells (and also on the probability that a place cell will recur from one session to the next), we not believe that these small effects or are of much interest or that they are likely to have impacted the outcome of our other analysis, for reasons discussed in the caption.

Importantly, there is poor information on how the ROIs are defined, what methods are used for segmentation and how imaging from different sessions are treated (same ROIs, different ROIs, etc…), and what signals are evaluated dF/F with/without denoising, models of spike inference, etc…

We have addressed this by specifying the requested information more clearly in the methods section (ll. 808-840; as noted therein, exact parameters for the analysis may be found in the scripts provided on Github). We have also updated Figure 2A (upper right quadrant) to more clearly illustrate an example of how cell contours align across sessions.

Reviewer #2 (Recommendations for the authors):Here are some detailed comments.1) Please add page and line numbers to ease review.

The revision is paginated and line numbered.

2) Given that several of the rats did the drug-free shock learning twice, how were these data analysed? Was only one of the sessions per rat chosen to contribute to the analysis, if yes, how?

Only one rat (the female shown in the bottom row of Figure 2B) received two drugfree shocks, and the second drug-free shock session was not included in the analysis. To avoid confusion, the second drug-free shock session is no longer shown in Figure 2 B (since no data from this animal was used after the extinction session that is still plotted in the Figure). We thank the reviewer for catching this.

3) Was there an influence of the order of sessions on the results (e.g. scopolamine during the first shock session, the second one, after a barrier session or not)?

A ranksum test yielded no significant effect of training session order on results within either the drug free or scopolamine shock conditions; this is now indicated by ‘ns’ labels underneath the horizontal axis in Figure 1H, right panel.

4) Any analysis of the number of trials, if comparing across sessions of different durations, should be done on a number of trials/minute to allow for cross-sessions comparisons (e.g. Sup Figure 2 A).

It is not clear to us why the referee is recommending this. The goal of beeline trial filtering is to assure that behavior sampling of the short path (total time spent in each bin) is as similar as possible across analyzed sessions; this mandates equalizing the TOTAL NUMBER of beeline trials per session, not trials per unit time. Hence, the primary reason for showing panel A in Supplement 2 to Figure 2 (formerly Sup Figure 2A to which the referee refers) is to show what pre-subsampled trial counts look like when they are NOT normalized by time, and thereby demonstrate why it was necessary to equalize trial counts by subsampling beeline trials. If the graph were normalized by time, then the information the graph is intended to convey would be lost.

5) Analysis of remapping: remapping is not the same as instability. It corresponds to a consistent activation of place fields in specific locations, that change (in a consistent manner) in a different condition, with the activity pattern, mostly reinstated when coming back to the original condition. Otherwise, changes in place cell firing that are not consistent or reproducible are considered drift or instability of the spatial firing. It is unclear if the current paradigm and analysis tests remapping and not simply instability.

It is unclear what remedy, if any, the referee is requesting to address this concern.

6) In the 'Scopolamine does not disrupt place cell responses to shock' section: individual place cells are considered to 'respond' to the shock if their activity increases during entry into the shock zone. However, place cells can also respond by decreasing their activity, or even changing their place field shape or location, but this is not analysed.

Prior to submitting the original manuscript we did perform an analysis of inhibitory responses to shock. The problem with such an analysis is that statistical power for detecting excitatory versus inhibitory responses is highly asymmetric; the majority of place cells have a low baseline spike rate at the shocked location, so an immense number of trials would be required to detect a shock-evoked decrease of the spike rate from the already low baseline rate. Consequently, very few cell can be categorized as inhibited by shock, but whether this reflects an actual paucity of shock inhibition or a statistical power problem is impossible to determine.

7) Given that place cells can modulate their firing spontaneously with time (mentioned by the authors), any analysis of place cells between two different conditions where time has elapsed should have as a control a similar comparison, without the parameter of interest being changed, but with approximately the same amount of time passed or at least number of sessions (as the authors do in one of their analyses).

Yes, we have controlled for this. The amount of time that elapses between sessions is identical for all analyses in Figure 2 (48 h) and very nearly so for all analyses in Figures 3 and 4 (mode of 8d).

8) Nomenclature: In general, it would be good to avoid the use of abbreviations as much as possible, including in the figures, to help readability. Also, the different terms used (mostly condition or session names) can be confusing and there may be errors in figures or legends that make the interpretation difficult. The use of pretraining, training, and post in some instances, but pre10 post 5 in others (within the same session), is also a bit confusing. Perhaps the 10 min part of the shock training sessions that is pre-shock should be called the normal session and then the shock session would be called the shock session (or barrier session), without lumping both in the same session. Also, there are 2 forms of training: training to choose the shortest path, then training of the shock – and then extinction of the shock learning. Perhaps the authors could find a more intuitive and consistent way to name their sessions. What is 'post-extinction' (e.g. in 'post-extinction CA1 population vectors)? Is it referring to the activity during the extinction sessions? Or during the sessions that followed the extinction sessions, which according to Figure 2B will always be a shock session? It seems that this is sometimes called post-training (e.g. Figure 1J) and sometimes post-extinction (e.g. Figure 1J, legend). Is the place field recurrence rate (Figure 2G) the same as place field recurrence ratio (in results)? Do subsampled and down sampled mean the same here?

We sympathize with the referee’s confusion over these issues and have endeavored to clarify them throughout the revised manuscript.

9) Related to this, in several instances the axis labels or legends of a figure are not self-sufficient and could be enhanced with more information about the measure that is shown (e.g.: Figure 1J: how is %baseline computed?).

This has been clarified in the caption for panel J of Figure 1 which now reads:

“J) Left: Extinction curves over the first 3 post-extinction sessions; number of rewards earned on the short path is measured as a percentage of baseline (48 h before training; thin lines: individual rats, thick lines: per session median).”

10) The graph in Figure 1B seems to show that after each shock session, the rats have 1 extinction session. But in the main text, it is said that 'rats continued to run standard sessions on the maze every 48h until they reached the avoidance extinction criterion", which seems to assume that this would take several sessions. It seems that the graph should be adapted to reflect how many sessions were run. It is not clear what happens after the barrier sessions – do the rats move on to the shock learning sessions immediately? Does the '3s' mean that they do instead of the barrier session 3 times, or maybe extinction sessions?

This has been clarified by adding the following to the caption for panel B of Figure 1:

“The number of sessions in ACQ and EXT boxes varied per rat, depending on how many sessions it took individual subjects to reach behavioral criteria.”

11) It is a bit unclear what the -48/+48 measure means on some of the figure axes and in text. Does that mean that what is compared is data from 4 days apart? Or just 2 days apart? If it's the latter, shouldn't the measure be 0-48h or -24/+24?

4 days apart. To clarify this, the label “time from training session” has been added to the horizontal axis of Figure 1H.

12) Rejecting specific trials from the analysis does not change the behaviour of the rats during data collection; it only makes the analysis more balanced. The sentence "This assured that rats were behaving as similarly as possible during data collection in all sessions" should be rewritten to reflect this.

The statement has been rewritten as follows (ll. 884-887):

“This assured that rats were behaving as similarly as possibly during analyzed data segments from all sessions.”

13) Figure 2E could show proportions out of the total number of putative pyramidal cells analysed, not just out of detected place cells, to see if the percentage of active place cells changes.

Percentage of cells meeting place cell criteria are now reported in Figure 2D (right graph) and 3C (middle graph).

14) Tuning curve 'sequences' (showing the tuning curves of all cells ordered according to peak firing) are used in many of the plots, but it is unclear what information to take from them. Reorganizing the figures to emphasize the quantification of the results might be useful and perhaps removing some of the sequence plots unless they convey a specific message or can be used as an example.

We have retained the heatmap plots because they show al data from individual cells and therefore provide readers with important validation of the size and quality of the dataset that was analyzed in the other summary graphs.

15) Related to this, most of the figures are really small. Focusing on the ones that provide quantitative results (i.e. less 'tuning curves' plots) and making those bigger would probably help the reader.

We believe that the sizing of our figures conforms with current practice for publishing calcium imaging data in peer reviewed journals, but we will be happy to consider any figure reformatting guidance from the editorial staff if the paper is accepted for publication.

16) Descriptions of some specific procedures could be added in the methods (listed in no specific order): a) How were the beeline trials selected?

This is explained in great detail in the Methods section (ll. 879-944).

b) How were putative spikes extracted/reconstructed from the calcium imaging signal (deconvolution?)?

This is now explained in the Methods (ll. 834-840).

c) How was the injection procedure performed, and where were the rats kept after it? Were there any observable effects on the behaviour?

Injection methods are now explained in the Methods (ll. 794-798). Behavior effects are detailed in Results (ll. 112-146) and Discussion (ll. 561-612).

d) what was the weight/age of the rats at the start of the experiment?This is now stated in the Methods (ll. 724-727).e) what is the statistical significance threshold chosen?

It is unclear to what analysis this question is referring.

f) how were the shocks delivered and calibrated? Did they not generate artefacts in the imaging?

Shock delivery apparatus and procedures are now explained in the Methods (ll. 731-737). Shocks did not generate artefacts in the imaging. Unlike electrophysiological recordings, calcium imaging is not susceptible to shock artefacts. The housing of the miniscope is made of non-conductive plastic and is therefore electrically insulated from the rat and the shock current.

17) The difference between past findings and the novelty of the present study could be explained better (for example, Sun et al. 2021 seems very relevant to the current study but is practically not discussed, also, Wu et al. (2017) [Hippocampal awake replay in fear memory retrieval. Nature Neuroscience 20, 571-580] and Girardeau et al. (2017) [Hippocampal awake replay in fear memory retrieval. Nature Neuroscience] seem like they should be discussed).

We have added references to the Wu et al. (2017) and Girardeau et al. (2017) studies in the context of our discussion about how shock-induced remapping affects the spatial distribution of place fields (l. 522-24). We have also added new references to recently published work by Wirtshafter and Disterhoft (2022) and Ormond et al. (2023) that has come out during the time we have been working on revisions.

18) How were the A>0.6 threshold and A<0.31 threshold chosen?

As explained in the Methods (ll. 873-877), these values were chosen because they were the cutoff thresholds for the top and bottom quartiles, respectively, of the percentage of trials on which rats chose the short path.

19) The authors have selected for analyses of the 'control' (non-scopolamine) group only rats that learn to avoid the shock zone, and for analyses of the 'test' (scopolamine) group the rats that do not learn. Possibly related to this, it seems that the different groups have different behaviours in the pre-scopolamine session. I believe this is not the right approach; instead, if anything, the groups that are going to be compared should be equalized in as many parameters as possible at least prior to the manipulation of interest. If anything, the variability within each group should be used to assess the correlation between remapping/ instability of place cells and performance.

We report an analysis of correlation between remapping measures and behavioral performance in Panel H of Supplement 8 to Figure 2.

Reviewer #3 (Recommendations for the authors):On the whole, I very much enjoyed reading this work, I found the results to be quite interesting and the presentation to be clear. I have only one question that can be addressed with further analysis. An essential claim of the manuscript is that the spatial codes measured during the drug-free and scopolamine shock sessions are equivalent in terms of within-session quality (and I appreciated all of the subsampling controls to match gross behavioral measures), yet I think this is only shown to be the case for 'place cells' which were preselected on the basis of various within-session measures. Can you confirm that: (1) the percentage of cells meeting place-cell criteria is equivalent across the two manipulations (not just the raw number of cells),

Percentage of cells meeting place cell criteria are now reported in Figure 2D (right graph) and 3C (middle graph).

and that (2) the equivalency of these measures holds when including all principal cells, not just those meeting (same-session) preselection criteria? I would be convinced about this claim if this were the case.

The within-session quality measures to which the referee is referring (place field width, spatial info, peak in-field firing rate, mean out-of-field firing rate) are measures that can only be applied to cells with verified spatial tuning, so we are unsure what the referee means by requesting that these measures be performed on “all principle cells…not just those meeting preselection criteria.”

[Editors’ note: what follows is the authors’ response to the second round of review.]Reviewer #2 (Recommendations for the authors):Hippocampal place cell remapping occurs with memory storage of aversive experiences.Tracking no: 29-05-2022-RA-eLife-80661R1The scientific approach and interpretation of this revised manuscript have been substantially updated (mostly in the introduction and discussion) and appear much more straightforward and supported by the results. The authors have answered most of my comments and added a behavioural-only control for the injection procedure, which is very useful (although the results for the control group might need some clarification, see comment below). Overall, I find the manuscript to be much improved and the results very interesting.Comments related to previous comments:Previous Comment 4: Normalizing trial counts: Currently, the data shown in Figure 2 – supplement 2 in the 'trn' group is taken from 10 minutes of behaviour, while data in the other groups are obtained from 15-minute sessions. Thus, the figure gives the impression of the trial number decreasing between the pre and trn sessions, but this is due (at least partially) to the shorter session duration for trn. This is why I was suggesting using trials/min in order to see if the rats still run fewer trials for an identical duration. If the authors prefer to keep the absolute number of trials as a measure, I suggest adding in the figure itself a clear mention of the duration of data included in each group to avoid misunderstandings.

As explained in our reply to the referee’s original comment, retaining the nonnormalized scale is essential for conveying the information provided by the figure; we agree with the referee’s concerns about lack of clarity and have made the recommended changes by adding data durations to x-axis labels of the figure, along with a clarification of what these labels mean in the caption.

Previous Comment 5 Analysis of remapping vs instability: I believe the authors have actually done an analysis that answers this, by computing the within-session correlation of rate maps (showing that the cells are still spatially stable) in Figure 2I, by showing that the spatial information is similar between the pre and trn sessions even in the scopolamine condition, together with computing the pre vs post correlations; thus no action is needed.

Acknowledged.

Previous Comment 6 Response to shock by increased firing vs decreased firing: My suggestion would be to detect all active place fields contacting the shock zone in the pre10 session part, and analyze their firing in the post5 or subsequent session part. Using an active place field should prevent the low baseline problem described by the authors.

It would certainly be possible to skirt the problem of asymmetrical statistical power (noted in our prior reply) by performing such an analysis, but it is unclear to us that the results of such an analysis would be germane to the questions we are asking in the present study about whether shock responses are predictive of remapping responses. Analyzing inhibitory responses by the proposed method would only be possible for a subset of cells selected for the property of being active in the shock zone, a property that itself might be confounded with the dependent variable of interest: remapping. Moreover, some kind of inclusion threshold on statistical power would have to be formulated and justified to select cells by this criterion (the referee does not propose suggestions for how to do this). For these reasons, it is our view that the only question to which the proposed analysis could provide a meaningful answer would be “are there any place cells that show evidence of being inhibited by shock?” That question, although interesting, is not primary to the scope and focus of the present study.

Previous Comment 16.e Choice of significance threshold: Sorry if the previous comment was unclear – I meant that the methods should specify which is the threshold for statistical significance chosen to be used throughout the manuscript to decide when a test result that produces a p-value is deemed significant vs not, e.g. 'tests were considered significant at p<0.05'.

We have addressed this point by locating every statement about “significance” in the Results and Methods sections where threshold information was not provided in the text or figures; threshold information has been added on lines 215, 216, 235, 236, 460.

New Comment 1 (NC1): Thank you for adding a behavioural control with injection of saline. I have some comments related to this new figure (Figure 1 – Supplementary figure 1):a) Is data in the 'Shock' session from the entire 15 min or only the first 10 min pre-shock? I recommend showing separately pre10 and post5 to get a better understanding of the effects.

We have clarified this by adding the duration of each session in minutes to the session labels above the plot in Figure 1 – supplementary figure 1B; the caption text has also been updated with the following text:

“shock session includes only data from the 10’ pre-shock period whereas all other days include data from the full 15’ session.”

b) If all the data were included: why is it that the saline rats do not seem to avoid the shock zone during the shock session (compare pre and shock data in B or C)? This should be discussed in the main text.

As noted above (see reply to ‘a’) shock session data only includes the pre-shock period, so there is no inconsistency to address here.

c) Separating the data in the graph would also allow us to replicate the effect seen in the implanted rats where scopolamine rats avoid the short path even before the shock session. Does this effect underlie an increase in impulsivity, or impairment of decision-making, possibly linked to a decrease in general attention? Are such side-effects of the scopolamine known? Perhaps these types of possible side effects of the scopolamine could be added to the relevant discussion (e.g. in around L 596).

We assume that when the referee says “scopolamine rats avoid the short path even before the shock session” they mean before the shock STIMULUS (not “session”). It is indeed the case that both implanted and non-implanted scopolamine rats show reduced preference for the short path prior to shock. “Separating the data in the graph” is not necessary to see this; in the existing version of Figure 1 – supplementary figure 1 (panels B and C) it can be seen that on the shock day, scopolamine rats show a greatly reduced preference for the short path compared with saline-injected rats. New statistics have been added to the figure caption to show that this effect is significant.

d) Ideally, some measure of general activity (average speed or total distance covered for the whole session, not just included trials) should be shown in the same way as the existing figures to further assess the effects of the scopolamine, injection procedures and shock on activity.

We have now included the measure of median running speed during beeline trials for this data as well in Supplement to Figure 1A (bottom) and the caption has been revised to reflect this (L 1076):

“A) Top: Mann-Whitney U tests found that 48 hour retention of avoidance behavior did not differ for males versus females in the drug-free or scopolamine shock conditions. Bottom: Median beeline running speeds per session.”

e) Why is the 'shock' session not called 'trn' as in Figure 2 Supplement 2?

We have changed the label from ‘shock’ to ‘trn’ for consistency with figures in the main text.

NC2: L 24, L 52 (and also L 503) "We conclude that place cell remapping occurs in response to motivationally significant (rather than neutral) stimuli that are remembered rather than merely perceived and forgotten"; "studies have shown that place cells remap in response to behaviorally significant events": this is partially correct, however, these sentences and the corresponding one in the discussion imply that only behaviourally significant events can cause place cell remapping. But place cells also remap in response to events of low/no behavioural significance, e.g. a change in the odour/colour of the environment (Anderson and Jeffery 2003, J Neurosci; Jeffery et al., 2003) or events of intermediary behavioural relevance like the introduction of barriers in the absence of a navigation task (Rivard et al., 2004, Journal of General Physiology, Muller and Kubie, 1987, J Neurosci). These studies generally observe partial remapping (of a sub-population of the place cells) while perhaps the remapping in the shock condition in the current study is more global (but to test this, additional analysis of place fields away from the shock zone would be necessary). For these reasons, I suggest reformulating/removing the references to "motivationally significant (rather than neutral) stimuli" to perhaps focus on the remembered vs non-remembered aspect. Perhaps the salience of an event influences the degree of remapping throughout the place cell population… but that hasn't really been demonstrated by the current study.

On lines 24 and 502, we have followed the referee’s suggestion by changing the wording to read:

“…remapping occurs selectively under conditions where a stimulus occurs and is subsequently remembered rather than forgotten…”

We have retained existing wording on line 52 in the citation of prior studies showing that “place cells remap in response to behaviorally significant events” because we believe this wording is accurate and it is clearly not intended as a statement about what is demonstrated in the current study. We agree with the referee’s point that “behavioral significance” may not be a core prerequisite for remapping, but the existing wording of the citation is fully compatible with this, especially since the Introduction explains that remapping can occur with the mere passage of time and that the rate of remapping may be proportional to the rate at which episodic memories are being stored.

NC3: L508 +: "Hippocampal place fields tend to be preferentially concentrated at locations where rodents spend large amounts of time […]" : this is not always the case (see for example Hok et al., 2007, J Neurosci; Pfieffer, 2022, J Neurosci), and might depend on specific task demands and behavioural stereotypy (reviewed in Nyberg et al., 2022). I suggest adding a clarification that the goal overrepresentation phenomenon is not ubiquitous and might depend on the task and navigational strategy.

“tend to be” has been replaced with “are sometimes observed to be”

NC4: L 536 'indicating that place fields shifted towards and away from the shocked location in equal proportions': could the authors point to a result showing this, instead of fields, for example, changing their size or firing rate?

To better support this conclusion, we have added another supplementary figure (Figure 3—figure supplement 4) to quantify the field shifts.